

# Evaluation of Southern Ocean cloud in the HadGEM3 general circulation model and MERRA-2 reanalysis using ship-based observations

Peter Kuma[1], Adrian J. McDonald[1], Olaf Morgenstern[2], Simon P. Alexander[3], John J. Cassano[4], Sally Garrett[5], Jamie Halla[5], Sean Hartery[1], Mike J. Harvey[2], Simon Parsons[1], Graeme Plank[1], Vidya Varma[2], and Jonny Williams[2]

[1]School of Physical and Chemical Sciences, University of Canterbury, Christchurch, New Zealand
[2]National Institute of Water and Atmospheric Research, Wellington, New Zealand
[3]Australian Antarctic Division, Kingston, Australia
[4]Cooperative Institute for Research in Environmental Sciences and Department of Atmospheric and Oceanic Sciences, University of Colorado, Boulder, Colorado, US
[5]New Zealand Defence Force, Wellington, New Zealand

**Correspondence:** Peter Kuma (pku33@uclive.ac.nz)

**Abstract.** Southern Ocean (SO) shortwave (SW) radiation biases are a common problem in contemporary general circulation models (GCMs), with most models exhibiting a tendency to absorb too much incoming SW radiation. These biases have been attributed to deficiencies in the representation of clouds during the austral summer months, either due to cloud cover or cloud optical thickness being too low. The problem has been the focus of many studies, most of which utilised satellite datasets for

5  model evaluation. We use multi-year ship based observations and the CERES spaceborne radiation budget measurements to contrast cloud representation and SW radiation in the atmospheric component Global Atmosphere (GA) version 7.0 and 7.1 of the HadGEM3 GCM and the MERRA-2 reanalysis. We find that MERRA-2 is biased in the opposite direction to GA (reflects too much SW radiation). In addition, MERRA-2 performs better in terms of absolute SW bias than nudged runs of GA7.0 and GA7.1 in the 60–70°S latitude band. GA7.1 reduces the SO SW radiation biases relative to GA7.0, but significant errors remain

10  at up to 20 $\mathrm{W m^{-2}}$ between 60 and 70°S in the austral summer months. Using ship-based ceilometer observations, we find low cloud below 2 km to be predominant in the Ross Sea and the Indian Ocean sector of the SO. Utilising a novel surface lidar simulator developed for this study, derived from an existing COSP-ACTSIM spaceborne lidar simulator, we find that GA7.0 and MERRA-2 both underestimate low cloud occurrence relative to the ship observations by 18–25% on average, though the cloud cover in MERRA-2 is closer to observations by about 7%. Based on radiosonde observations, we also find the low cloud

15  to be strongly linked to boundary-layer atmospheric stability and the sea surface temperature. GA7.0 and MERRA-2 agree well with observations in terms of boundary-layer stability, suggesting that subgrid-scale parametrisations do not generate enough cloud in response to the thermodynamic profile of the atmosphere and the surface temperature. Our analysis shows that MERRA-2 has a much greater proportion of cloud liquid water in the SO in January than GA7.0, a likely key contributor to the difference in SW radiation. We show that boundary-layer stability and relative humidity fields are very similar in GA7.0 and



MERRA-2, and unlikely to be the cause of the different cloud representation, suggesting that subgrid-scale parametrisations are responsible for the difference between the models.

# 1   Introduction

Clouds are considered one of the largest sources of uncertainty in estimating global climate sensitivity (Boucher et al., 2013; Flato et al., 2013; Bony et al., 2015). Clouds over oceans are especially important for determining the radiation budget due to the low albedo of the sea surface compared to land. Over the Southern Ocean (SO), cloud cover is very high at over 80%, with boundary-layer clouds being particularly common (Mace et al., 2009). Excess downward shortwave (SW) radiation in general circulation models (GCMs), with a bias over the SO of up to 30 $\mathrm{Wm}^{-2}$, is a problem well-documented by Trenberth and Fasullo (2010) and Hyder et al. (2018), and has been the subject of many studies. Bodas-Salcedo et al. (2014) evaluated the SW bias in a number of GCMs and found that a strong SW bias is a very common feature, leading to increased sea surface temperature (SST) in the SO and corresponding biases in the storm track position. Trenberth and Fasullo (2010) note that a poor representation of clouds might lead to unrealistic climate change projections in the Southern Hemisphere. The SW bias has also been linked to large-scale model problems such as the double-Intertropical Convergence Zone (Hwang and Frierson, 2013), biases in the position of the midlatitude jet (Ceppi et al., 2012) and errors in the meridional energy transport (Mason et al., 2014). Bodas-Salcedo et al. (2012) studied the SO SW bias in the context of the Global Atmosphere (GA) 2.0 and 3.0 models and found that mid-topped and stratocumulus clouds are the dominant contributors to the bias.

Due to its extent and magnitude, the SW radiation bias is believed to limit accuracy of the models, especially for modelling the Southern Hemisphere climate. A model based on the Hadley Centre Global Environmental Model version 3 (HadGEM3) is currently used in New Zealand for assessing future climate (Williams et al., 2016). In this paper we evaluate two versions of the atmospheric component of HadGEM3, GA7.0 and GA7.1 (Walters et al., 2017) using observations collected in the SO on a number of voyages. The main objective of this study is to evaluate SO cloud in GA7.0 and GA7.1 based on ship-based remote sensing and in situ observations. Ship-based atmospheric observations in the SO provide a unique view of the atmosphere not available via any other means. Boundary layer observations by satellite instruments are limited by the presence of an almost continuous cloud cover, potentially obscuring the view of low level clouds. The frequently used active instruments CloudSat (Stephens et al., 2002) and Cloud–Aerosol Lidar and Infrared Pathfinder Satellite Observation (CALIPSO) (Winker et al., 2010) are both of limited use when observing low level, thick or multi-layer cloud: CloudSat is affected by surface clutter below approximately 1.2 km (Marchand et al., 2008) and the CALIPSO lidar signal cannot pass through thick cloud. Likewise, passive instruments and datasets such as the Moderate Resolution Imaging Spectroradiometer (MODIS) (Salomonson et al., 2002) and the International Satellite Cloud Climatology Project (ISCCP) (Rossow and Schiffer, 1999) can only observe radiation scattered or emitted from the cloud top of optically thick clouds. Therefore, one can accurately identify the cloud top height or cloud top pressure with satellite instruments, but not always the cloud base height (CBH) or the vertical profile of cloud, although there has been some recent progress on deriving CBH statistically from CALIPSO measurements (Mülmenstädt et al., 2018). Ship-based measurements therefore provide valuable extra information.



Multiple explanations of the SW radiation bias have been proposed: cloud underestimation in the cold sectors of cyclones (Bodas-Salcedo et al., 2014), cloud–aerosol interaction (Vergara-Temprado et al., 2018), cloud homogeneity representation (Loveridge and Davies, 2018), lack of supercooled liquid (Kay et al., 2016; Bodas-Salcedo et al., 2016) and the "too few, too bright" problem (Nam et al., 2012; Klein et al., 2013; Wall et al., 2017). Each model can exhibit the bias for a different set of

reasons, and results from one model evaluation therefore do not necessarily explain biases in all other models (Mason et al., 2015). The use of SO voyage data for atmospheric model evaluation is not new, and has recently been used by Sato et al. (2018) to evaluate the impact of SO radiosonde observations on the accuracy of weather forecasting models. Klekociuk et al. (2018) contrasted SO cloud observations with the ECMWF Interim reanalysis (ERA-Interim) and the Antarctic Mesoscale Prediction System–Weather Research and Forecasting Model (AMPS-WRF), and found that these models underestimate the coverage

of the predominantly low cloud. Protat et al. (2017) compared ship-based 95 GHz cloud radar measurements at 43–48°S in March 2015 with the Australian Community Climate and Earth-System Simulator (ACCESS) NWP model, a model related to HadGEM3, and found low cloud peaking at 80% cloud cover, which was underestimated in the model. It was also more spread out vertically (especially due to "multilayer" situations defined as co-occurrence of cloud below and above 3 km) and more likely to have intermediate cloud fraction in the model. Previous studies have documented that supercooled liquid is

often present in the SO cloud in the austral summer months (Morrison et al., 2011; Huang et al., 2012; Chubb et al., 2013; Huang et al., 2016; Bodas-Salcedo et al., 2016; Jolly et al., 2018) and is linked to SO SW radiation biases in GCMs, which underestimate the amount of supercooled liquid in clouds in favour of ice. Warm clouds generally reflect more SW radiation than cold clouds containing the same amount of water (Vergara-Temprado et al., 2018). In particular, Kay et al. (2016) reported a successful reduction of SO absorbed SW radiation in the Community Atmosphere Model version 5 (CAM5) by tuning the

shallow convection ice detrainment temperature and thereby increasing the amount of supercooled liquid cloud.

Two common techniques used for model cloud evaluation have been cloud regimes (Williams and Webb, 2009; Haynes et al., 2011; Mason et al., 2014, 2015; McDonald et al., 2016; Jin et al., 2017; McDonald and Parsons, 2018; Schuddeboom et al., 2018, 2019) and cyclone compositing (Bodas-Salcedo et al., 2012; Williams et al., 2013; Bodas-Salcedo et al., 2014, 2016; Williams and Bodas-Salcedo, 2017), both of which link the SW radiation bias to specific cloud regimes and cyclone sectors.

Jakob (2003) discusses different methods of cloud evaluation. We use simple statistical techniques, rather than sophisticated classification or machine learning algorithms, the advantage of which is easier interpretation for the purpose of model development. We first assess the magnitude of the Top of Atmosphere (TOA) SO SW radiation bias in the GA7.0 and GA7.1 models and the Modern-Era Retrospective analysis for Research and Applications, version 2 (MERRA-2) reanalysis with respect to the Clouds and the Earth's Radiant Energy System (CERES) Energy Balanced and Filled (EBAF) and CERES Synoptic (SYN)

products. This allows us to identify the underlying magnitude of the SW bias and how this might change based on the ship track sampling pattern. We then evaluate cloud occurrence in GA7.0 and MERRA-2 relative to the SO ceilometer observations and compare SO radiosonde observations with pseudo-radiosonde profiles derived from the models. Lastly, we look at zonal plots of potential temperature, humidity, cloud liquid and ice content in GA7.0 and MERRA-2 to show how these models differ in their atmospheric stability and representation of clouds. Our aim is to identify how differences between GA7.0 and MERRA-2

can explain the TOA SW bias, assuming misrepresentation of clouds is the major contributor to the bias.





## 2 Methods

We used an observational dataset of ceilometer and radiosonde data comprising multiple SO voyages (Figure 1), GA7.0 and GA7.1 atmospheric model simulations (Walters et al., 2017) and the MERRA-2 reanalysis (Gelaro et al., 2017). Later in the text, we will refer to GA7.0, GA7.1 and MERRA-2 together as "the models", even though MERRA-2 is more specifically a
reanalysis. CERES satellite observations (Wielicki et al., 1996) were also used as a reference for TOA SW radiation and an National Snow and Ice Data Center (NSIDC) satellite-based dataset (Maslanik and Stroeve, 1999) was used as an auxiliary dataset for identifying sea ice. CFMIP Observation Simulator Package (COSP) (Bodas-Salcedo et al., 2011), a set of instrument simulators developed by the Cloud Feedback Model Intercomparison Project (CFMIP), was extended with a surface lidar simulator and used to produce virtual lidar measurements from model fields (Kuma et al., 2019). Resampling, noise reduction
and cloud detection were performed on observational and (where applicable) model lidar data in a consistent way to reduce structural uncertainty (see Section 2.4). The schematic in Figure 2 shows the processing pipeline utilised in this study.

### 2.1 Datasets

#### 2.1.1 HadGEM3

HadGEM3 (Walters et al., 2017) is a general circulation model developed by the UK Met Office and the Unified Model Part-
nership. It is used either in a free-running mode or "nudging" (Telford et al., 2008) – relaxing winds and potential temperature towards the ERA-Interim reanalysis (Dee et al., 2011). The Met Office Global Atmosphere 7.0 and 7.1 (GA7.0 and GA7.1, respectively) is the atmospheric component of HadGEM3 (Walters et al., 2017).

The following runs were used in our analysis:

– 1980–89 run of GA7.0 free-running ("GA7.0U/1980-1989").

– 2007 run of GA7.0 nudged ("GA7.0N/2007").

– 2007 run of GA7.1 nudged ("GA7.1N/2007").

The model runs used the HadISST sea surface temperature dataset (Rayner et al., 2003) as lateral boundary conditions. The nudged simulations represent atmospheric dynamics as determined by observations. In the free-running model, atmospheric dynamics can only be compared statistically with observations or reanalyses. The model was run on a $1.875° \times 1.25°$ (longitude
$\times$ latitude) "N96" resolution grid, which corresponds to a horizontal resolution of about $100 \times 140$ km at 60°S and 85 vertical levels. The model output was provided as instantaneous fields sampled every 6 hours. Limited data availability meant that no nudged runs were available for the period of 2015–2018 when the ship observations are available. Therefore, we used the decadal free-running simulation to compare cloud representation statistically.



### 2.1.2 MERRA-2

Modern-Era Retrospective analysis for Research and Applications (MERRA-2) is a reanalysis provided by the NASA Global Modelling and Assimilation Office (Gelaro et al., 2017). The reanalysis was chosen for its contrasting results of TOA shortwave radiation bias in the SO compared to GA7.0 and GA7.1. Its bias is positive rather than negative, when CERES is used as a reference.

We used the following products (Bosilovich et al., 2015):

– 1-hourly average Radiation Diagnostics (product "M2T1NXRAD.5.12.4")

– 3-hourly instantaneous Assimilated Meteorological Fields (product "M2I3NVASM.5.12.4")

– 1-hourly instantaneous Single-Level Diagnostics (product "M2I1NXASM.5.12.4")

– 3-hourly average Assimilated Meteorological Fields (product "M2T3NVASM.5.12.4")

– 1-hourly average Single Level Diagnostics (product "M2T1NXSLV.5.12.4")

We used the "Radiation Diagnostics" in TOA SW radiation evaluation (Section 3.1), the instantaneous "Assimilate Meteorological Fields" and "Single-Level Diagnostics" products to generate simulated ceilometer profiles and pseudo-radiosoundings (Section 3.2 and 3.3), and the average "Assimilate Meteorological Fields" and "Single-Level Diagnostics" to generate zonal plane plots of thermodynamic and cloud fields (Section 3.4). Before running the COSP simulator, we downsampled the grid resolution to a $1° \times 1°$ grid in order to reduce the computational demands. The 4-dimensional MERRA-2 fields were provided on pressure and model levels. For our analysis we chose to use the model-level products (72 levels) due to their higher vertical resolution compared to pressure-level products.

### 2.1.3 Ship observations

We use ship-based ceilometer and radiosonde observations made in the SO on 5 separate voyages between 2015 and 2018 (Table 1 and Figure 1):[1]

– 2015 TAN1502 voyage of the NIWA ship RV *Tangaroa* from Wellington, New Zealand to the Ross Sea.

– 2015–2016 voyages (V1–V3) of the Australian Antarctic Division (AAD) icebreaker *Aurora Australis* from Hobart, Australia to Mawson, Davis, Casey and Macquarie Island ("AA15")

– 2016 Royal New Zealand Navy (RNZN) ship HMNZS *Wellington* voyages ("HMNZSW16").

– 2017 NBP1704 voyage of the NSF icebreaker RV *Nathaniel B. Palmer* from Lyttelton, New Zealand to the Ross Sea.

---

[1]The voyage name pattern is a 2–6 character ship name followed by a 2 digit year and a 2 digit sequence number. TANxxxx and NBPxxxx are official voyage names, while HMNZSW16 and AA15 are names made for the purpose of this study.



– 2018 TAN1802 voyage of RV *Tangaroa* from Wellington to the Ross Sea (Hartery et al., 2019).

Together, these voyages cover latitudes between 41 and 78°S and the months of November to June inclusive. A total of 298 days of observations were collected. Geographically, the voyages mostly cover the Ross Sea sector of the SO, with only AA15 covering the Indian Ocean sector (Figure 1). This sampling emphasises the Ross Sea sector over other parts of the SO,

although the SO SW radiation bias appears largely zonally symmetric (Section 3.1), with a notable exception of the eastern side of the Antarctic Peninsula, as is the atmospheric circulation in the SO (Jones and Simmonds, 1993; Sinclair, 1994, 1995; Simmonds and Keay, 2000; Simmonds et al., 2003; Simmonds, 2003; Hoskins and Hodges, 2005; Hodges et al., 2011), which should allow these results to be extrapolated over the whole of SO at the affected latitudes. Figure 1 shows the tracks of the voyages used in this study. The voyage observations were performed using a range of instruments (described below). Table 2

details which instruments were deployed on each voyage.

The primary instruments were the Lufft CHM 15k and Vaisala CL51 ceilometers. A ceilometer is an instrument which typically uses a single-wavelength laser to emit pulses vertically into the atmosphere and measures subsequent backscatter resolved on a large number of vertical levels based on the timing of the retrieved signal (Emeis, 2010). Depending on the wavelength, the emitted signal interacts with cloud droplets, ice crystals and precipitation by Mie scattering, and to a lesser

extent with aerosol and atmospheric gases by Rayleigh scattering (Bohren and Huffman, 2008). The signal is quickly attenuated in thick cloud and therefore it is normally not possible to observe mid and high level parts of such a cloud, or a multi-layer cloud. The main derived quantity determined from the backscatter is CBH, but it is also possible to apply a cloud detection algorithm to determine cloud occurrence by height. The range-normalised signal is affected by noise which increases with the square of range. A major source of noise is solar radiation which causes a diurnal variation in noise levels (Kotthaus et al.,

2016). Due to noise and signal attenuation, the cloud profile retrieved by a ceilometer does not directly reveal the cloud liquid and ice mixing ratios in an atmospheric model output, and a lidar simulator has to be used to account for these effects (Chepfer et al., 2008). The Lufft CHM 15k ceilometer operates in the near-infrared spectrum at 1064 nm, measuring lidar backscatter up to a maximum height of 15 km, producing 1024 regularly spaced bins (about 15 m resolution). The sampling rate of the instrument is 2 s. The Vaisala CL51 ceilometer operates in the near-infrared spectrum at 910 nm. The sampling rate of the

instrument is 2 s and range is 7.7 km, producing 770 regularly spaced bins (10 m resolution).

On the TAN1802 voyage we used the iMet-1 ABx radiosondes, measuring pressure, air temperature, relative humidity and GNSS coordinates of the sonde (from which wind speed and direction are derived). The sondes were launched three times per day at about 8:00, 12:00 and 20:00 UTC on 100 g Kaymont weather balloons. They reached a typical altitude of 10–20 km, and then terminated by balloon burst or loss of radio communication. We used 10 s resolution profiles generated by the

vendor-supplied iMetOS-II control software for further processing. We also had access to automatic weather station (AWS) data from some of the voyages (RV *Tangaroa* and RV *Nathaniel B. Palmer*). These included variables such as air temperature, pressure, sea surface temperature, wind speed and wind direction. Voyage track coordinates were obtained from the ships' Global Navigation Satellite System (GNSS) receivers.



### 2.1.4 CERES

The Clouds and the Earth's Radiant Energy System (CERES) is a set of low Earth orbit (LEO) satellite instruments and a dataset of SW and longwave (LW) radiation observations (Loeb et al., 2018; Doelling et al., 2016). The CERES instruments (called FM1 to FM6) provide a continuous record of observations since the first deployment on the Tropical Rainfall Measuring Mission (TRMM) satellite in 1997 (Simpson et al., 1996), and have been flown on Terra, Aqua (Parkinson, 2003), the Suomi NPOESS Preparatory Project (Suomi NPP) and Joint Polar Satellite System-1 (JPSS-1) (Goldberg et al., 2013) satellites since. Currently CERES is considered the best available global Earth radiation datasets, and is often used as the primary dataset for GCM tuning and validation (Schmidt et al., 2017; Hourdin et al., 2017). We used the following CERES products in our analysis:

- CERES SYN1deg-Day Edition 4A (configuration code 401405) product of daily average radiation ("CERES SYN").

- CERES EBAF-TOA Edition 4.0 (CERES_EBAF_Ed4.0) product of monthly energy-balanced average radiation ("CERES EBAF").

Due to the sun-synchronous orbits of the LEO satellite platforms, the Flight Model (FM) instruments of CERES do not capture the full diurnal variation of radiation. The EBAF and and SYN1deg products are adjusted for diurnal variation by using 1-hourly geostationary satellite observations between $60°$S and $60°$N, and use an algorithm to account for changing solar zenith angle and diurnal land heating. The CERES EBAF-TOA Edition 4.0 product is a Level 3B product, which means it has been globally balanced by ocean heat measurements using the Argo network (Roemmich and Team, 2009).

### 2.1.5 NSIDC sea ice concentration

We used the Near-Real-Time Defense Meteorological Satellite Program (DMPS) Special Sensor Microwave Imager/Sounder (SSMIS) Daily Polar Gridded Sea Ice Concentrations, Version 1 product (NSIDC-0081) (Maslanik and Stroeve, 1999) provided by the National Snow and Ice Data Center (NSIDC) to classify observations into those affected and unaffected by sea ice. The sea ice concentration product has a resolution of $25 \times 25$ km. We used a cutoff value of 15% of sea ice concentration for the binary classification of sea ice, in line with previous studies (Comiso and Nishio, 2008).

### 2.2 Domains

Because our observational dataset does not span the entire geographical area of the SO or all months of the year, and the atmospheric conditions in the SO are geographically variable, we subset our datasets into a number of geographical regions by latitude and time periods by season. The three geographical regions identified are $55$–$60°$S, $60$–$65°$S and $65$–$70°$S and the time periods are austral summer, months December–January–February (DJF) and autumn months March–April–May (MAM). Although we have a substantial quantity of data taken at latitudes south of $70°$S, we do not use them here, as they would likely be affected by circulation induced by land near the Ross Sea (Coggins et al., 2014), and therefore may not be representative of the SO in general. This decision builds on the analysis detailed in Jolly et al. (2018) which shows a significant gradient in cloud



properties between the Ross Ice Shelf and the Ross Sea and strong influences associated with synoptic conditions. Likewise, we would have to exclude land areas, which have very different atmospheric climatologies.

There is likely temporal variability present within the austral summer and austral autumn periods, but we decided to limit the number of temporal classes to maintain a reasonable quantity of observations in each class in this analysis. The magnitude of

the SO TOA SW radiation bias is primarily modulated by incoming solar radiation, which is the highest in the austral summer period. The voyages do not uniformly cover all geographical regions or time periods, with the largest number of observations in the Ross Sea sector south of New Zealand (TAN1802, TAN1502, HMNZSW16, NBP1704), followed by the Indian Ocean sector south of Western Australia (AA15). Temporally, the voyage observations mostly cover summer to late summer/autumn months of the year. When subsetting model data, we sample along voyage tracks (geographically and temporally), and in the

case of the free-running GA7.0 simulation, we compare 10 years of model data statistically, and the same time period relative to the start of the year.

## 2.3 COSP simulator

COSP was originally developed as a satellite simulator package whose aim is to produce virtual satellite (and more recently ground-based) observations from atmospheric model fields in order to improve comparisons of model output with observations

(Bodas-Salcedo et al., 2011). This approach is required because physical quantities derived from satellite observations generally do not directly correspond to model fields. COSP accounts for the limited view of the satellite instrument by calculating radiative transfer through the atmosphere, i.e. attenuation by hydrometeors and air molecules and backscattering. COSP comprises multiple instrument simulators, such as MODIS, ISCCP, MISR, CALIPSO and CloudSat. It has been used extensively by previous studies of model cloud, for example by Kay et al. (2012), Franklin et al. (2013), Klein et al. (2013), Williams and

Bodas-Salcedo (2017), Jin et al. (2017) , and Schuddeboom et al. (2018). COSP is planned to be used in the upcoming Coupled Model Intercomparison Project Phase 6 (CMIP6) (Webb et al., 2017).

For our analysis, we have developed a ground-based lidar simulator based on the COSP CALIPSO spaceborne lidar simulator (Chiriaco et al., 2006) (see the Code and data availability section at the end of the document). This required reversing of the vertical layers, as the surface lidar looks from the surface up rather than down from space to the surface, and changing the

radiation wavelength affecting Mie scattering by cloud droplets and Rayleigh scattering by air molecules. In this paper we present only a brief description of the surface lidar simulator, with a more complete description planned in an upcoming paper. These changes will be contributed to the upstream COSP project, or made publicly available, so that the scientific community can reuse the surface lidar simulator in the future.

The recently introduced COSP version 2 (Swales et al., 2018) added support for a surface lidar simulator, although we believe

our implementation, developed before COSPv2 was available, is more complete in the present context due to its treatment of Mie scattering at wavelengths other than 532 nm (the wavelength of the CALIPSO lidar). Previously, a surface lidar simulator based on COSP has been used by Chiriaco et al. (2018) and Bastin et al. (2018). A ground-based radar simulator in COSP has also recently been implemented (Zhang et al., 2018). The surface lidar simulator takes model cloud liquid and ice mixing ratios, cloud fraction and thermodynamic profiles as the input, and calculates vertical profiles of attenuated backscatter. This



can be done either by running the simulator "online" within the model code or "offline" on the model output. We used the offline approach in our analysis.

## 2.4 Lidar data processing

Lidar data in this study came from two different instruments: Lufft CHM 15k and Vaisala CL51 ceilometers and the lidar
simulator. These instruments use different output formats, wavelengths, sampling rates and range bins, as previously noted. Backscatter and derived fields such as CBH are provided in the firmware generated data products, but the backscatter is uncalibrated and the derived fields such as cloud detection are based on instrument-dependent algorithms. Therefore, we performed consistent subsampling, noise reduction and cloud detection on data from both instruments, and applied the same methods to the lidar simulator output. As part of the processing we developed a publicly available tool called cl2nc ("CL to
NetCDF") for converting the Vaisala CL51 ceilometer data format to NetCDF (see the Code and data availability section at the end of the document).

### 2.4.1 Calibration

The backscatter profiles produced by the Lufft CHM 15k and Vaisala CL51 ceilometers are not calibrated to physical units, even though they are expressed in $\mathrm{m}^{-1}\mathrm{sr}^{-1}$. To calibrate these backscatter fields we used the method described by O'Connor
et al. (2004). This method uses the lidar ratio (LR) to calculate a calibration factor based on a known value of the LR in fully scattering cloudy scenes, such as thick stratocumulus clouds, which are common over the SO. We applied this technique by using visually identified scenes and choosing a calibration factor which achieves the known value. Due to the nature of the conditions (LR can be highly variable even in thick cloud scenes), the calibration is likely accurate to only about 50% of the backscatter value. We do not expect this to have a serious impact on the accuracy of cloud detection completed in this study,
largely because the predominantly low cloud tends to cause backscatter orders of magnitude greater than clear air, and because of the very large differences in cloud occurrence between the observations and models. Kotthaus et al. (2016) provide a detailed description of backscatter retrieval by Vaisala ceilometers.

### 2.4.2 Subsampling, noise removal and cloud detection

In order to simplify further processing and increase the signal-to-noise ratio, we subsampled the ceilometer observations at a
sampling rate of 5 minutes by averaging multiple profiles, and vertically averaging on regularly spaced 50 m bins. We expect that in most cases cloud was almost constant on this time and vertical scale, and therefore we were not averaging together different cloud types or clear and cloudy profiles. At the same time as subsampling, we performed noise removal by estimating the noise distribution (mean and standard deviation) based on returns in the uppermost range bins (i.e. 300 samples over 5 min when sampling rate was 2 s), and subtracting the range-scaled noise mean from the backscatter. We then used the range-scaled
noise standard deviation ($\sigma$) for cloud detection: a bin was considered cloudy if the calibrated backscatter minus $3\sigma$ exceeded $20\times10^{-6}\ \mathrm{m}^{-1}\mathrm{sr}^{-1}$. This threshold was chosen subjectively so that cloud was visually well separated from other features, such



as boundary-layer aerosol and noise on backscatter profile plots. The same threshold was used on both the observations and output from the COSP surface lidar simulator and thus should cause little bias.

### 2.4.3 Model lidar data processing

We used the same sampling rate (5 min) and model levels as range bins on the surface lidar simulator output. For each vertical
profile we used model data at the same location as the ship and the same time relative to the start of the year. Model data were selected using nearest-neighbour interpolation. The model resolution is lower than the distance travelled by the ship in 5 minutes, therefore the same model data were used multiple times to generate consecutive profiles. However, we also used the SCOPS (Webb et al., 2001) subcolumn generator included in COSP to generate 10 random samples of cloud for each profile based on cloud fraction and the maximum/random cloud overlap assumption (Bodas-Salcedo, 2010). The lidar simulator does
not generate noise, and therefore we did not perform any noise removal on the simulated profiles, but we used the same threshold of $20 \times 10^{-6}$ m$^{-1}$sr$^{-1}$ and vertical bins of 50 m for detecting cloud (as used on the observations). For the MERRA-2 cloud occurrence analysis, we applied the lidar simulator on the 3-hourly instantaneous Assimilated Meteorological Fields (M2I3NVASM.5.12.4) product subsampled to a $1 \times 1$ degree global horizontal grid.

### 2.5 SST lifting level

In our analysis we used a metric "SST lifting level" (SLL) derived from SST and boundary-layer atmospheric potential temperature (measured by radiosondes or simulated by a model). We define SLL as the level to which an air parcel with the same temperature as SST, rising from the sea surface, would rise adiabatically by buoyancy. That is, it is the level closest to the surface at which potential temperature is equal to SST, provided the air parcel is permitted to rise to this level by buoyancy (otherwise the air parcel does not rise and SLL is 0 m). This metric is applicable in sea ice-free conditions in the SO, when cold
Antarctic air is warmed by the open sea surface and is lifted by buoyancy until it reaches a limit imposed by the atmospheric stability of the atmosphere. Together with the lifting condensation level (LCL) we found SLL to be a useful metric for evaluation of boundary-layer CBH. Apart from SST and LCL, we also evaluate cloud with respect to lower tropospheric stability (LTS) (Klein and Hartmann, 1993).

## 3 Results

### 3.1 Shortwave radiation balance

Figure 3 shows reflected TOA SW radiation in CERES, GA7.0, GA7.1 and MERRA-2. We present this panel plot in order to evaluate how well GA7.0, GA7.1 and MERRA-2 are performing in terms of the SW radiation bias in the SO relative to CERES. This analysis assumes that CERES is a good observational reference, although it is affected by biases of lower order of magnitude (Loeb et al., 2018). The plots reveal a predominantly zonally symmetric pattern of reflectivity in the SO
on the yearly (Figure 3a–d) and monthly scales (Figure 3e–h), with more variable patterns in the tropics related to regions





of upwelling and downwelling. We chose 19 January 2007 as a representative day in January to show the daily pattern[2]. On the daily scale (Figure 3i–l; 19 January 2007) the patterns are closely linked to synoptic features, with close inspection displaying particularly large differences in the TOA SW radiation near the Antarctic Peninsula. The region on the eastern side of the Antarctic Peninsula shows a greater reflectivity in CERES (Figure 3e), but not in any of the models (Figure 3f–h). The

zonal symmetry of the annual and monthly means (Figure 3a–h) suggests that there is not a significant need for subsetting by longitude, and that latitude averages can be very useful in identifying the key features of the SW radiation biases. The synoptic features are generally well-correlated between CERES and the models (Figure 3i–l), which is expected in nudged model runs and reanalyses. The highest reflectivity is generally associated with frontal regions and extratropical/polar cyclones, although cloud-associated reflectivity is present throughout the SO. Examination shows that MERRA-2 has greater upwelling TOA SW

radiation on all three time scales presented here. Considering that cloud is the dominant factor affecting SW radiation in the SO, this can only be associated with either cloud cover which is too high, or clouds which are too bright, and our analysis of cloud occurrence (Section 3.2) supports the latter. GA7.0 and GA7.1 are less reflective than CERES between $60°S$ and $70°S$ (Figure 3m, n), with some individual cloud systems being too bright (Figure 3j, k). MERRA-2 is also much more reflective on the January monthly mean at all latitudes between $55°S$ and $70°S$ (Figure 3o). The opposing sign of the SO SW radiation bias

in GA7.0 and GA7.1 compared to MERRA-2 suggests that contrasting the two models could be useful in uncovering the cause of the SO SW radiation biases.

    Figure 4 shows line plots of zonal mean reflected SW radiation and bias relative to CERES by month in multiple latitude bands between 55 and $70°S$, with the southernmost band $65–70°S$ limited to $180–80°W$ to avoid land areas in Antarctica. The annual cycle follows the expected cyclical pattern modulated by varying incoming solar radiation with maxima of reflected

radiation in December and maxima of bias in December and January. The Antarctic sea ice extent, at its minimum in February and peaking in September, is also likely a secondary modulating factor at higher latitudes. The models represent the cyclical pattern well, but differ substantially during periods of peak incoming solar radiation. Inspection of the GA7.0 model (Figure 4b, f, j, n) shows a largely negative bias in the SO, increasing with latitude and reaching -38 $Wm^{-2}$ between 65 and $70°S$ in January. Between 50 and $55°S$ (Figure 4b) however, the bias is positive at its peak, reaching 5 $Wm^{-2}$ and overall is close to

zero throughout the year. This is important because previous studies of SO cloud often do not discern different latitudes, partly due to the limited availability of surface and in situ cloud observations in the SO. These panels also justify why it is important to do spatial subsetting by latitude when analysing the SO SW bias in models. The GA7.1 model (Figure 4c, g, k, o) exhibits lower bias than GA7.0 at all latitude bands except for $50–55°S$ (Figure 4c), where the positive bias is greater, peaking at 10 $Wm^{-2}$ and is fairly constant throughout the year. The likely explanation for this feature is that GA7.1 is reflecting more SW

radiation in the SO, which reduces the bias where it is negative, but increases the bias where it is already positive. Overall, GA7.1 improved the peak bias from -38 $Wm^{-2}$ to -20 $Wm^{-2}$ at $65–70°S$ in January. MERRA-2 displays a clearly different bias than GA7.0 and GA7.1 (d, h, l, p). The SW bias is consistently positive, i.e. too much SW radiation is reflected at all latitudes between 50 and $70°S$. The absolute value of the bias is also lower than in GA7.0 and GA7.1 between 60 and $70°S$ (15 $Wm^{-2}$ in MERRA-2 vs. -20 $Wm^{-2}$ in GA7.0). Therefore, the MERRA-2 results are valuable for contrasting with GA7.0 and

---

[2]A choice made for convenience during the analysis due to overlap with existing Transpose-AMIP HadGEM2-A hindcasts.





GA7.1. In the low latitude SO (50–60°S), however, MERRA-2 performs more poorly than GA7.0 and GA7.1, showing bias of about 30 $\mathrm{Wm^{-2}}$.

To summarise, we find that GA7.1 is an improvement over GA7.0 with respect to the SW radiation bias in the SO, and MERRA-2 is superior to GA7.1 at latitudes poleward of 60°S. However, due to compensating biases commonly present in the models it might not be a superior representation of reality. Schuddeboom et al. (2019) evaluate compensating errors in the GA7.1 model.

## 3.2 Cloud occurrence in model and observations

To understand how clouds contribute to the SW bias, we examine cloud cover and cloud occurrence as a function of height in the models and observations. Figure 5 shows cloud occurrence profiles derived from ceilometer observations on different voyages and GA7.0U and MERRA-2 model output derived via the COSP surface lidar simulator, as a function of latitude and season. The seasons cover the austral summer and late summer/autumn months. The comparison with GA7.0 is completed statistically, i.e. 10 individual years of free-running GA7.0 simulation data between 1980 and 1989 are compared with observations and reanalysis between 2015 and 2018. Most notably, the observed cloud cover is consistently very high in the observations (80–100%) for all periods and latitude bands examined and greater than 90% in most of these subsets. This differs substantially from the modelled cloud cover (derived via the surface lidar simulator), which ranges between 17 and 92% in GA7.0, and is about 25% lower than observations across the subsets. Cloud cover in MERRA-2 is also generally lower than observations, but higher than GA7.0, spanning 49–95%. Our analysis therefore shows that cloud cover is underestimated in both GA7.0 and MERRA-2 in the geographical regions and seasons evaluated here. This shows a similar bias to previous analysis by Schuddeboom et al. (2018) who compared COSP derived cloud cover from the GA7.0 model with MODIS satellite observations and found much higher cloud cover in the observations. Due to the high zonal symmetry of the SW radiation in the SO, shown in Figure 3, and the magnitude of the cloud cover bias in the model, these results are likely representative of the whole SO. Examination of the vertical distributions in Figure 5 shows that the observations indicate a strong predominance of cloud below 2 km, peaking below 1 km, including a substantial amount of surface-level fog in some subsets. In contrast, GA7.0 simulates cloud at a higher altitude, peaking at about 1 km. Further analysis shows that the MERRA-2 vertical distribution appears more consistent with the observations than GA7.0, often peaking below 1 km, but overall having lower cloud cover at the peak altitude than the observations.

Figure 6 shows the model subsets of Figure 5 as points by their cloud cover bias relative to observations. It can be seen that GA7.0U underestimates cloud cover by about 25% and MERRA-2 by 18% when non-weighted averages are considered, and both models underestimate this amount by about 18% when weighted averages are considered. This difference is caused mostly by an outlier: AA15 DJF 65–70°S (28.2 days), which exhibits 80% cloud cover in observations and 90% in GA7.0U. Neither the averages nor weighted averages, however, should be accepted uncritically, as they group together different latitudes and statistically correlated weather situations. The same weather can persist for several days, and therefore measurements taken during a continous period of time are statistically correlated, whereas measurements on different voyages represent statistically independent samples.



Due to the nature of the lidar measurements, mid to high level cloud may be obscured by low level cloud, because the laser signal is quickly attenuated by thick cloud. Therefore, the lack of cloud above 2 km in the plots does not imply that there is no cloud at these heights. The results demonstrate the value of surface cloud measurements in the SO relative to satellite measurements such as CloudSat and CALIPSO, which would likely provide a biased sample of these clouds because of "ground clutter" and obscuring higher level cloud, respectively (Alexander and Protat, 2018).

## 3.3 Radiosonde observations

Radiosonde observations were performed on the TAN1802 and NBP1704 voyages. Temperature, pressure, relative humidity and GNSS coordinates (from which wind speed and direction are derived) were retrieved to altitudes of 10–20 km, terminated by a loss of radio communication or balloon burst. We use these data to evaluate boundary-layer properties and correlate them with cloud as observed by a ceilometer. We compare the observations with "pseudo-radiosonde" profiles extracted from model fields at the same location and time of the year. Figure 7 shows the relationship between CBH and the minimum of SSL and LCL ("min{SLL,LCL}") as a scatter plot based on a merged dataset from TAN1802 and NBP1704 voyages, and the corresponding points from GA7.0 and MERRA-2. We choose to evaluate min{SLL,LCL} as a predictor instead of either SSL or LCL individually for the following reasons. This relationship becomes quite notable when examining the individual voyage radiosonde profiles (not presented here). If SLL is higher than LCL, an air parcel warmed by the sea surface rises by buoyancy past LCL, at which point water vapour starts to condensate (assuming enough cloud condensation nuclei are present at 100% saturation), forming cloud with CBH equal to LCL. If SLL is lower than LCL, the air parcel rises to SLL, where air lifted from the sea surface eventually accumulates, potentially forming cloud if enough moisture is transported from the sea surface. A large fraction of the observed points (OBS) in Figure 7a lies close to the origin, which suggests that near zero min{SLL,LCL} is a good indicator of fog or very low cloud. The models do not seem to represent this relationship well. The histogram in Figure 7a shows that about 40% of observed profiles have CBH within 100 m of min{SLL,LCL}, while only about 20% of MERRA-2 profiles and 15% of GA7.0U/1980-89 profiles do. Using SSL or LCL as a predictor for CBH individually resulted in a weaker relationship than min{SLL,LCL} (not presented here). Figure 7b shows the same points as a function of LTS, defined as the difference between potential temperature at 700 hPa and sea level pressure (Klein and Hartmann, 1993), and used in previous studies (Williams et al., 2006; Franklin et al., 2013; Williams et al., 2013; Naud et al., 2014). LTS does not display a good predictive ability for CBH in this dataset, with the exception of very stable profiles (LTS > 15 K), when observed CBH was below 250 m in all but one case.

Figure 8 shows the distribution of SSL as derived from radiosonde observations and model fields, and scatter plots of CBH vs. min{SLL,LCL} as in Figure 7. The purpose of the panel plot is to evaluate the relationship between local boundary-layer thermodynamics and cloud occurrence. In the absence of a synoptic-scale forcing and geographical features, one can expect clouds in the boundary layer to be well correlated with the local thermodynamic profile in the boundary layer. Due to the very persistent cloud cover observed in the SO in summer months (close to 100%), as shown by the cloud occurrence analysis in Figure 5, we might expect that conditions in the SO are such that an almost continuous cloud formation takes place or that cloud persists even in the absence of synoptic forcing. We hypothesise that the models underestimate cloud cover in these quiescent





conditions. As can be seen in the scatter plots in Figure 8, there is a strong correspondence between min{LCL,SLL} and CBH in cases where there is no sea ice. Because of the observed close link between SLL and CBH, we examined whether the models may be misrepresenting SLL. As can be seen in the SLL vertical distribution panels (Figure 8a–f, m–r), there is no substantial difference between the models and radiosonde observations in non-sea ice cases, in which SLL has a plausible effect on cloud,

even though GA7.0 simulates a slightly higher SLL than observed. We conclude that SLL difference is likely not a cause for the underestimated cloud cover in the models. SLL is a function of SST and the boundary layer potential temperature profile, we therefore expect both fields to be well simulated in the boundary layer of the models, and by ruling out this potential cause, the alternative explanation – that subgrid-scale model processes are an important factor in the underestimation of cloud cover – is more likely.

## 3.4    Zonal plane comparison of GA7.0 and MERRA-2

In order to better understand the differences in the SW bias between GA7.0 and MERRA-2, we inspect zonal plane plots of cloud occurrence and thermodynamic fields of both models in January and on a specific day (Figure 9). The GA7.0 model is a nudged run, which ensures a general correspondence between synoptic features in the model, the reality and the MERRA-2 reanalysis, i.e. both columns of Figure 9 show the same synoptic features. The figure shows monthly and daily average cloud

liquid and ice mixing ratio contours (a monthly average in January 2007 and a daily average on 19 January 2007, respectively) plotted over two different backgrounds – potential temperature and relative humidity. The daily average plot (Figure 9c, d) shows a very pronounced difference between the cloud liquid amount between the two models, with MERRA-2 simulating a much greater amount of cloud liquid. In contrast, GA7.0 simulates clouds with ice, which are absent in MERRA-2 at the chosen contour levels. The liquid content is generally concentrated near SLL as observed in Figure 8, and therefore at the top of

the surface coupled boundary layer, whereas the ice content in GA7.0 generally has significantly greater vertical extent. These differences are also present in the monthly average (Figure 9a, b). Relatively small differences in the background potential temperature and SLL between the two models fields suggest that the cloud differences are not explained by these fields. Moreover, the fields appear fairly consistent between the models, suggesting that the synoptic state of the atmosphere is not responsible for the cloud differences. There is, however, a pronounced difference in relative humidity between GA7.0 and

MERRA-2 in the mid to high troposphere, quite clearly visible on the monthly average plots (Figure 9e, f). This bias does not seem to be present in the boundary layer and it is therefore not a likely explanation for the cloud bias. We should note that it is not obvious from this analysis whether GA7.0 or MERRA-2 are closer to reality, even though larger amount of simulated cloud is preferable with respect to reducing the model SW radiation biases. Model cloud liquid and ice which is more spread out horizontally, while holding the same total amount of water, would increase the cloud cover a reduce the cloud opacity, and

lead to a better correspondence with our observations.





## 4 Discussion

The TOA SW radiation assessment showed that models exhibit monthly average biases of up to -38 $\mathrm{Wm^{-2}}$ (GA7.0, 65–70°S in January), and that these biases have a significant latitudinal dependency, leading to opposing signs of the bias between different latitude bands. This conclusion is also supported by Schuddeboom et al. (2019), who observed opposing sign of the

SW cloud radiative effect (CRE) south and north of 55°S in GA7.1. We found GA7.0/GA7.1 and MERRA-2 to be biased in the opposite direction, with GA reflecting too little SW radiation in the high latitude SO, while MERRA-2 reflects too much SW radiation in the SO. Consistent with the maximum of incoming solar radiation, January was found to be the month with the greatest absolute bias in models. For this reason, improving model cloud biases in austral summer months is more important than in other months with respect to the SW radiation bias. Cloud representation differences are expected to be the

strongest factor in modulating the TOA SW radiation, and this can happen either via cloud cover or cloud opacity effects, or both simultaneously. Therefore, we conclude that GA7.0/7.1 simulates too little cloud cover, but we cannot conclude whether it is too opaque or too transparent, and MERRA-2 simulates too little cloud cover and too opaque cloud. The cloud occurrence analysis revealed close to 100% cloud cover as measured by a ceilometer on a number of voyages. This seems to be the case across different latitudes in the austral summer and autumn, even though the results are limited by region (the Ross Sea and

Indian Ocean sectors) and the relatively brief passage of the ships through some of these regions. The 2016–2018 voyages may have been affected by the unusually low sea ice extent (discussed below), which can have a significant effect on cloud (Frey et al., 2018; Taylor et al., 2015). We found GA7.0 underestimates total cloud cover by a relatively large amount (nearly 25%), with MERRA-2 underestimating total cloud cover by a lesser extent. Combined with the overestimation of the TOA SW radiation, we concluded that MERRA-2 must be overestimating cloud opacity to an extent which overcompensates for the lack

of cloud cover. We cannot make the same conclusion about GA7.0, but it seems plausible that strongly underestimated cloud cover alone can explain the TOA SW radiation bias relative to CERES.

During the TAN1802 voyage we found a notable correspondence between CBH, SLL and LCL. Boundary layer thermodynamics, determining the lifting levels, is a plausible driver of cloud formation in the absence of other forcing. We examined SLL in models and radiosonde observations, and found differences which are likely too small to explain the cloud occurrence

differences between the models and ceilometer observations. Bodas-Salcedo et al. (2012), in their analysis of an earlier version of the GA model (GA3.0) using cyclone composites also noted that biases in thermodynamics are not likely to explain the SW radiation bias, but may still play a significant role. The presence of positive TOA SW radiation bias in the SO between 50 and 55°S in GA7.1, which has an opposing sign to the bias in the high latitude SO, is important because it places a limit on the applicability of other studies which used SO observational data from regions north of 55°S (Lang et al., 2018).

It is interesting to contrast our results with previous studies which used cyclone compositing for the TOA SW radiation bias evaluation in GCMs. We cannot make substantial conclusions from our results on how much of the model bias is attributable to cyclones. It appears, however, that the cloud cover and cloud liquid and ice mixing ratio bias in GA7.0 is systematic rather than isolated to cyclonic activities due to its relative consistency across spatiotemporal subsets in the high latitude SO. This does not rule out even greater biases related to cyclonic sectors. Specifically, Bodas-Salcedo et al. (2014) evaluated a large



set of models, including HadGEM2-A, a predecessor model to HadGEM3, likely affected by similar biases, and found that about 80% of grid cells south of 55°S could be classified as affected by a cyclone, and that these grid cells were responsible for the majority of the total SW radiation bias. Moreover, their cyclone compositing showed that the bias in HadGEM2-A was largely negative in the cold quadrants, and near zero in the warm quadrants. Their results also indicate a strong contrast in SW

bias south and north of 55°S, similar to the result we found in GA7.0 and GA7.1. We think these results can be reconciled with our study by assuming that the model has a particular difficulty in representing cloud in situations when near-surface air temperature is lower than the SST. In these regions the heat flux is from the ocean to the atmosphere is positive, which in the austral summer predominantly occur south of 55°S and in the cold sectors of cyclones. The cloud representation when near-surface air temperature is greater than SST is relatively more accurate, this case occurring predominantly north of 55°S

and in the warm sector of cyclones. To evaluate the viability of this explanation we plotted the daily average SW radiation bias in the GA7.0N/2007 grid cells as a function of near-surface air temperature and near-surface relative humidity between 40°S and 70°S on 19 January 2007 (Figure 10). The grid cells with strong negative bias are visibly clustered between -2 and +2 °C, whereas at higher temperatures the bias tends to be more equally distributed between positive and negative values. This suggests a possible explanation that subzero air mass advecting from Antarctica or from sea ice covered areas over warm water

could be inducing convection and steam fog or low cloud, and this process is not well represented by the model. Therefore the cloud biases in HadGEM2-A and HadGEM3 may not be linked to cyclonic activity as such, but secondarily through their impact on near-surface air temperature and its difference from SST.

Supercooled liquid was not a focus of this study, but we can note a number of things. Previous studies have documented that supercooled liquid is often present in the SO cloud in summer months. We cannot add to these findings with our observations,

although preliminary analysis of a polarising lidar (Sigma Space MiniMPL) profiles from the TAN1802 voyage suggests supercooled liquid was commonly present in the ubiquitous stratocumulus cloud. The side-by-side comparison of cloud liquid and ice mixing ratios on the zonal plane (Figure 9) suggests that models can differ significantly in their representation of cloud phase, with GA7.0 having less supercooled liquid than MERRA-2 (in January). Notwithstanding the cloud phase, the major problem of both models appears to be the lack of cloud cover compared to observations. If cloud cover is increased in the

model, the cloud opacity may also need to be lowered to obtain a good match of TOA SW radiation with CERES observations. We know that MERRA-2 overestimates cloud opacity, and GA7 may also be overestimating cloud opacity in order to partially compensate for the lack of cloud cover.

In our results there is some indication that sea ice has an impact on the cloud base height (Figure 8x), but it is not easily separated from the possible effect of the geographical location (high latitude Ross Sea region) and the time of the year (MAM).

The modulating effect of sea ice on cloud in the SO has previously been shown by Listowski et al. (2018) and there is an apparent difference in cloud between the Ross Sea and Ross Ice Shelf as shown by Jolly et al. (2018), with cloud over the ice shelf having smaller cloud cover, a greater amount of altostratus cloud and a smaller amount of deep convective cloud. The sea ice and ice shelves block transport of heat and moisture to the atmosphere. Their low thermal conductivity and high albedo mean the surface can cool to very low temperature and thus have an effect on the radiation balance of the atmosphere. We did

not focus on sea ice conditions, since one can expect the effect of cloud biases on the SW radiation bias over sea ice to be





small – the ice surface is already highly reflective in the SW, and the presence of cloud has little impact on the grid cell SW reflectivity (the SW albedo of cloud is similar to sea ice, depending on the sea ice concentration).

The Antarctic sea ice extent has undergone a rapid decrease starting in the spring of 2016 after about a decade of slightly increasing extent (Turner et al., 2017; Stuecker et al., 2017; Doddridge and Marshall, 2017; Kusahara et al., 2018; Schlosser et al., 2018; Ludescher et al., 2018). The sea ice extent due to this decrease was found to be the lowest on observational record since 1979, and the Ross Sea was particularly affected by this anomaly. The unusually low sea ice extent likely affected atmospheric observations made on the voyages presented in this study, e.g. the TAN1802 voyage in February and March 2018 to the Ross Sea experienced no sea ice during the entire voyage. Because sea ice is an important factor influencing the atmospheric boundary-layer stability and radiation balance, a significant secondary effect on cloud cover, cloud phase and opacity is expected. Sea ice is, however, not expected to be responsible for the SO SW radiation bias in models, because the bias is present even when sea ice concentration is prescribed from satellite observations. In our analysis, this may have an effect on comparison of cloud occurrence in the free-running GA7.0U relative to observations taken between 2016 and 2018. The representation of cloud in the nudged run and MERRA-2, however, should be comparable with observations without being affected by the sea ice anomaly due to having prescribed sea ice based on the satellite record.

## 5 Conclusions

We analysed 4 years of observational SO ship data, and contrasted them with a decade of free-running GA7.0 simulation, one year of nudged and free-running GA7.0 and GA7.1 simulation, and the MERRA-2 reanalysis. We used satellite observations of the Earth radiation budget to assess the TOA SW radiation bias in the SO in the three models. We examined the total cloud cover and vertical distribution of cloud as measured by ceilometers and simulated by a ceilometer simulator based on the model data. We also compared SO radiosonde observations from two voyages with virtual radiosonde profiles from the models in order to assess boundary-layer stability and the correlation between cloud base and atmospheric lifting levels. We also compared model fields of cloud liquid and ice content, potential temperature and relative humidity in a zonal plane analysis across the SO in order to contrast cloud and thermodynamics simulated by GA7.0 and MERRA-2.

Despite improvements, the SO SW radiation bias remains significant in the GA7.1 atmospheric model. SO ship-based lidar and radiosonde observations are a valuable tool for model cloud evaluation, considering the amount of low cloud in this region which is likely poorly sampled by satellite instruments due to possible obscuration by higher overlapping cloud. The main findings of this study are that multi-year ship-based observations:

- corroborate satellite-based evidence of underestimated cloud cover, with both GA7.0 and MERRA-2 underestimating cloud cover by 18–25%,

- show that low cloud below 2 km is almost continuous in the SO in summer months in sea ice-free conditions,

- indicate that boundary-layer thermodynamics is a strong driver of cloud in the SO, but this relationship is not well represented in the models,





– suggest that subgrid-scale processes in situations when near-surface atmospheric temperature is lower or close to SST are responsible for the cloud misrepresentation.

Future studies of SO cloud representation in the GA model could focus on specific details of the model subgrid-scale cloud processes, and how their tuning impacts cloud occurrence distributions compared to the ship observations. The stark difference between GA7.0 and MERRA-2 cloud liquid and ice content also remains to be explained, and could provide valuable insight for improving the SO SW radiation bias in the model and the reanalysis.

*Code and data availability.* The original COSP version 1 simulator is open source and available publicly at https://github.com/CFMIP/COSPv1. The modified COSP version 1 simulator including the ground-based lidar simulator used in this study is open source and available publicly at https://github.com/peterkuma/COSPv1. The cl2nc software for converting Vaisala CL51 data to NetCDF is available at https://github.com/peterkuma/cl2nc. The CERES EBAF and SYN1deg products are available publicly from the CERES website: https://ceres.larc.nasa.gov/. The Neal-Real-Time DMPS SSMIS Daily Polar Gridded Sea Ice Concentrations product is available publicly from the NSIDC website: https://nsidc.org/data/nsidc-0081. The Hadley Centre Sea Ice and Sea Surface Temperature data set (HadISST) is available publicly from the Met Office website: https://www.metoffice.gov.uk/hadobs/hadisst/. The MERRA-2 data are available publicly from the MERRA-2 website: https://gmao.gsfc.nasa.gov/reanalysis/MERRA-2/. The ship-based observations dataset is available on request from the authors.

*Author contributions.* Peter Kuma participated on methodology development, voyage observations, data analysis, writing and reviewing of the manuscript. Adrian McDonald participated on conceptualisation, funding acquisition, methodology development, voyage observations, data analysis, writing and reviewing of the manuscript. Olaf Morgenstern participated on model development, methodology development, data analysis, writing and reviewing of the manuscript. Simon Alexander, John Cassano, Jamie Halla, Sean Hartery, Sally Garrett, Mike Harvey, Simon Parsons, and Graeme Plank participated on voyage observations and reviewing of the manuscript. Vidya Varma and Jonny Williams participated on model development and reviewing of the manuscript.

*Competing interests.* The authors declare that they have no conflict of interest.

*Acknowledgements.* We would like to thank everyone who participated on obtaining the Southern Ocean voyage observations, especially Kelly Schick and Peter Guest for performing ceilometer and radiosonde measurements on RV *Nathaniel B. Palmer*; the Royal New Zealand Navy for ceilometer and radar measurements on HMNZS *Wellington*; Alex Schuddeboom for deployment of instruments on RV *Nathaniel B. Palmer*, the crew of the TAN1502, *Aurora Australis* V1–V3 2015/16, HMNZS *Wellington*, NBP1704 and TAN1802 voyages. Logistical and technical support for the ceilometer observations made aboard *Aurora Australis* during the summer of 2015/16 were provided as part of the Australian Antarctic Science project 4292. We acknowledge the Met Office for use of the MetUM, and for providing the HadGEM3 model. We acknowledge NASA-GMAO and ECMWF for the MERRA-2 and ERA-Interim reanalyses, respectively. In this analysis we





used publicly available satellite datasets provided by NASA and NSIDC. The CERES data were obtained from the NASA Langley Research
Center CERES ordering tool (https://ceres.larc.nasa.gov). We wish to acknowledge the contribution of the NeSI high-performance computing
facilities to the results of this research. New Zealand's national facilities are provided by the NZ eScience Infrastructure and funded jointly
by NeSI's collaborator institutions and through the Ministry of Business, Innovation & Employment's Research Infrastructure programme

5   (https://www.nesi.org.nz). We would like to acknowledge the financial support that made this work possible provided by the Deep South
National Science Challenge via the "Clouds and Aerosols" project. We acknowledge the software tools Python, R (R Core Team, 2018),
numpy (Oliphant, 2006), scipy (Jones et al., 2001–), matplotlib (Hunter, 2007), Climate Data Operators (CDO) (Schulzweida, 2018) and
parallel (Tange et al., 2011), which we used in our data analysis.





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





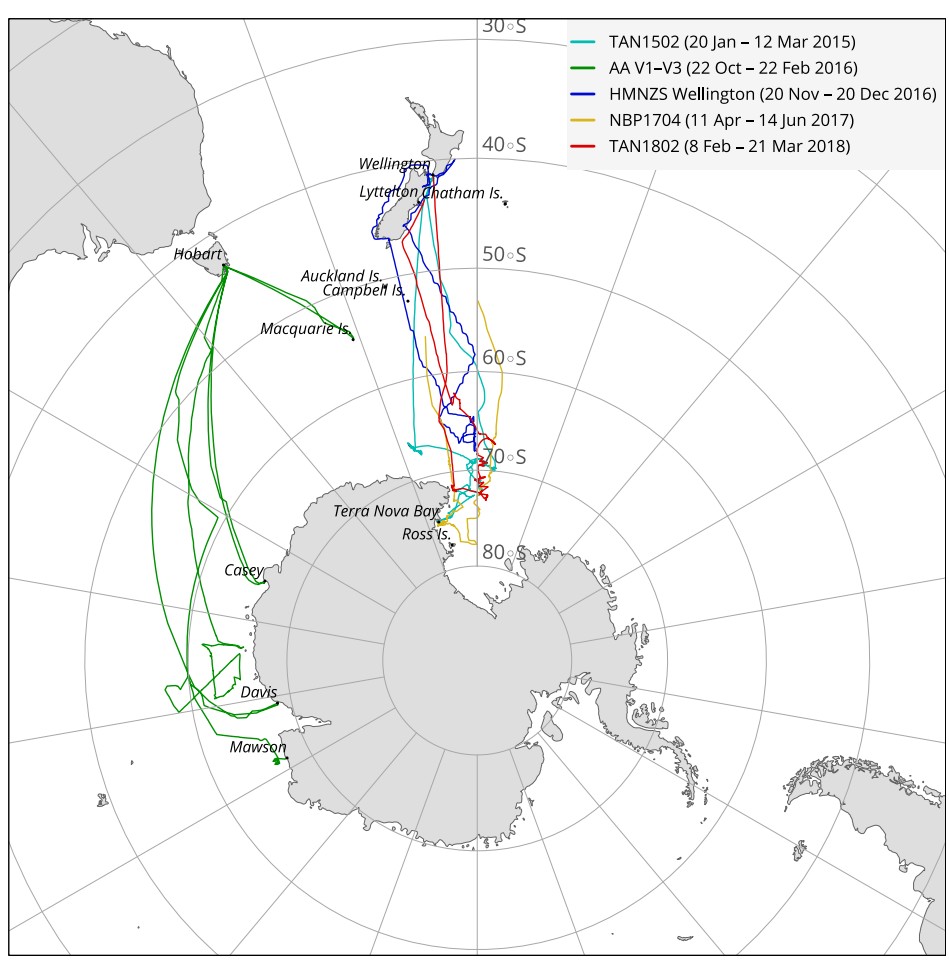

**Figure 1.** Map showing tracks of voyages used in this study. The ship observational dataset comprises 5 voyages between 2015 and 2018, spanning months from November to June and latitudes between 40°S and 78°S, of which data between 50°S and 70°S are used in this study.





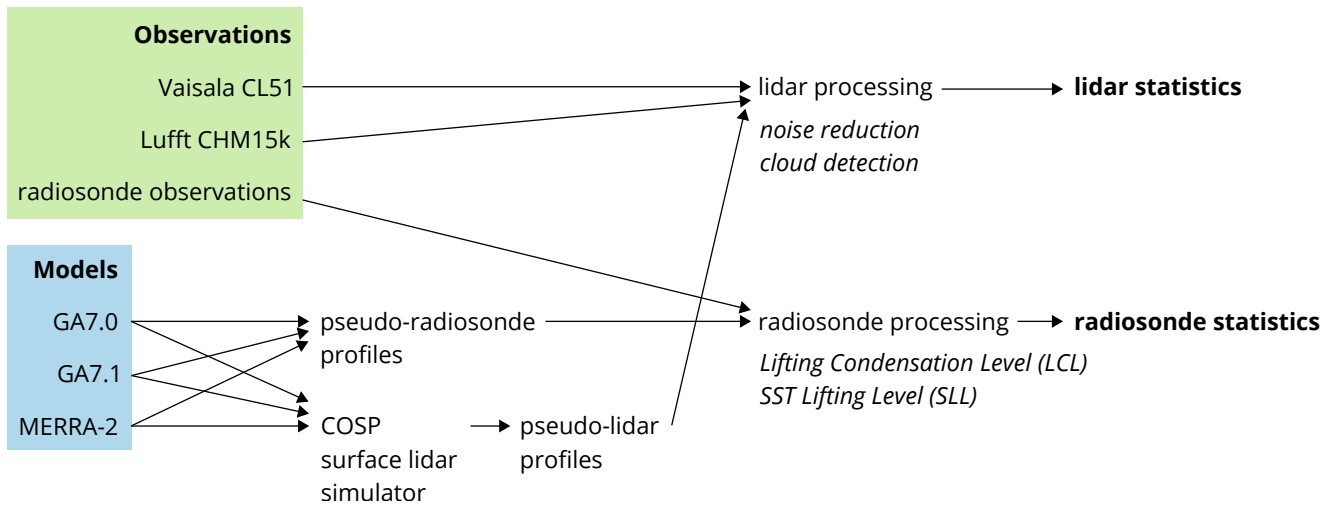

**Figure 2.** Schematic of the processing pipeline utilised in this study to produce lidar and radiosonde statistics from observations and model data.



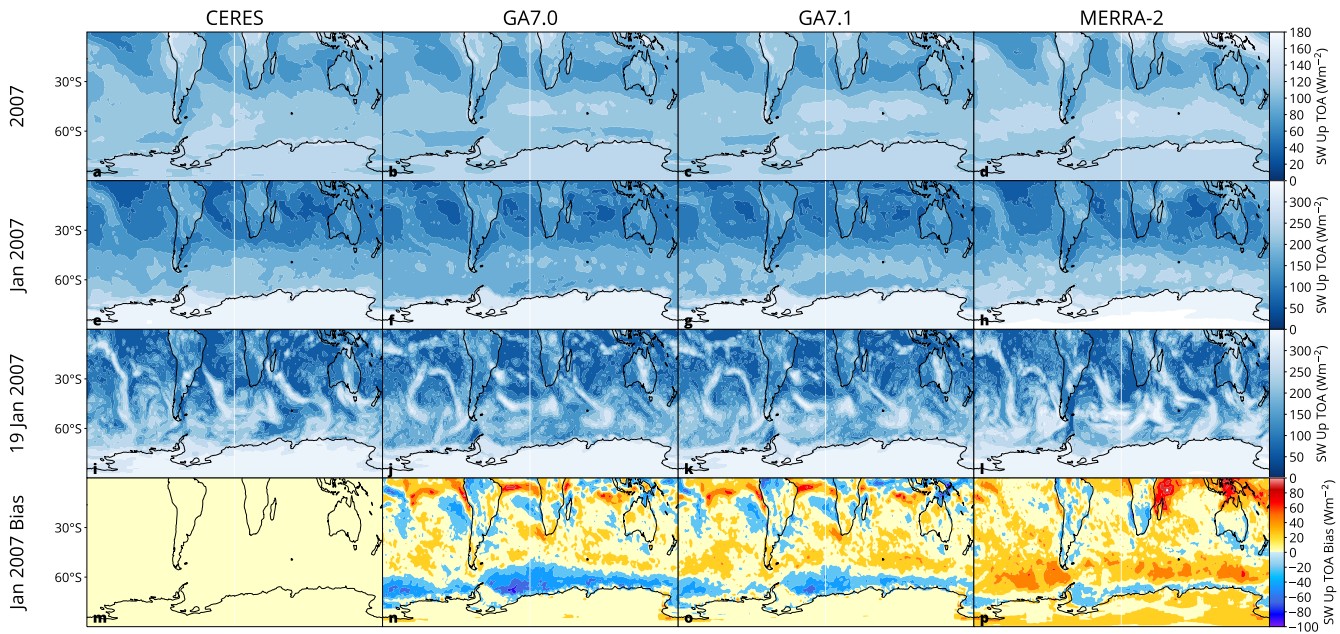

**Figure 3.** Geographical distribution of the TOA SW upwelling radiation in CERES and multiple models. The plots show global all sky SW radiation as a yearly (2007; **a–d**), monthly (January 2007; **e–h**), daily (19 January 2007; **i–l**) average and monthly average bias relative to CERES (January 2007; **n–p**). Highlighted are multiple latitude bands (55, 60, 65, 70°S). In **n–p**, positive (red) values indicate that the model overestimates reflected SW radiation, while negative (blue) values indicate that the model underestimates reflected SW radiation relative to CERES.



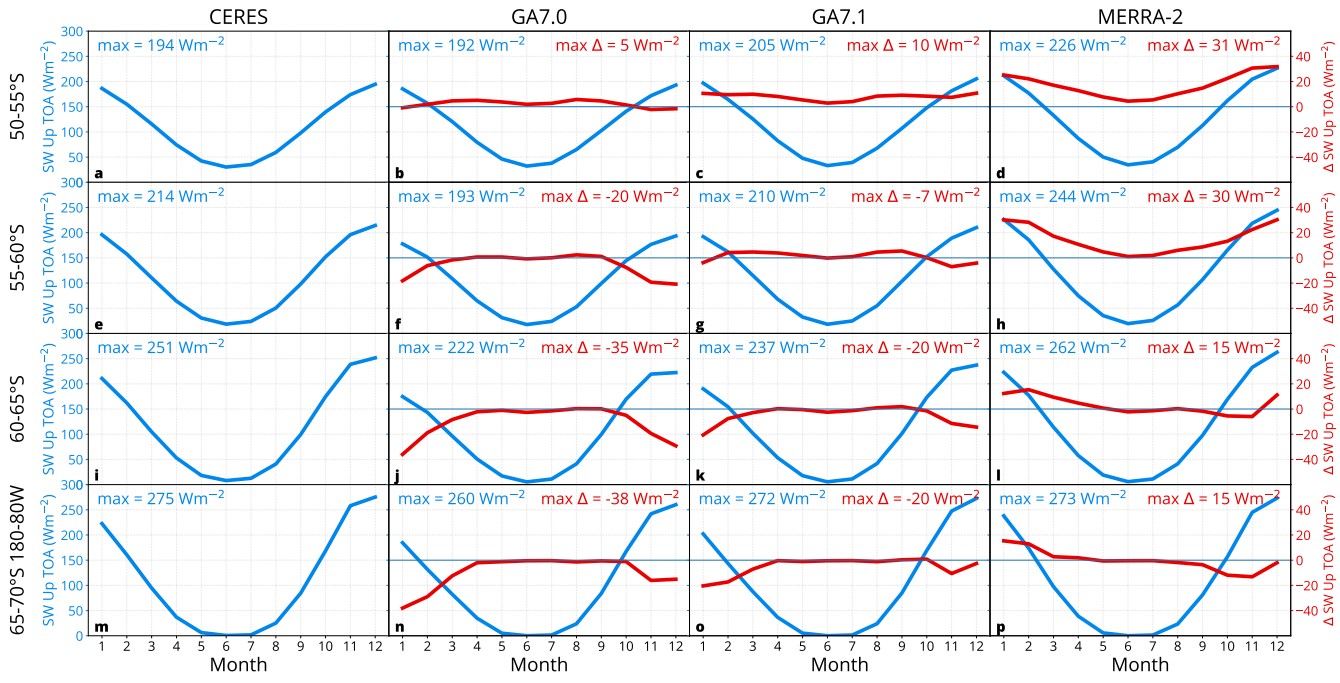

**Figure 4.** Zonal means of the TOA SW upwelling radiation in CERES and multiple models during the year 2007 in several latitude bands. The plots show time series of monthly zonal mean TOA SW upwelling radiation (blue) and its difference relative to CERES (red) as a function of month. Shown are also the maxima of the SW radiation and its difference from CERES.





**Figure 5.** Cloud occurrence frequency as a function of height derived from ceilometer backscatter (OBS) and model fields (GA7.0U and MERRA-2). The observational and model data were subsetted by latitude and season (DJF/December-January-February, MAM/March-April-May) along the voyage track. The numbers at the top of each panel show total (vertically integrated) cloud cover and the number of days the ship spent passing through the spatiotemporal subset. A 1-$\sigma$ confidence band calculated from a set of 10 years of the free-running GA7.0 simulation is indicated by a semi-transparent red band. The height in the plots is limited to 6 km. There was no significant amount of cloud detected above this level.



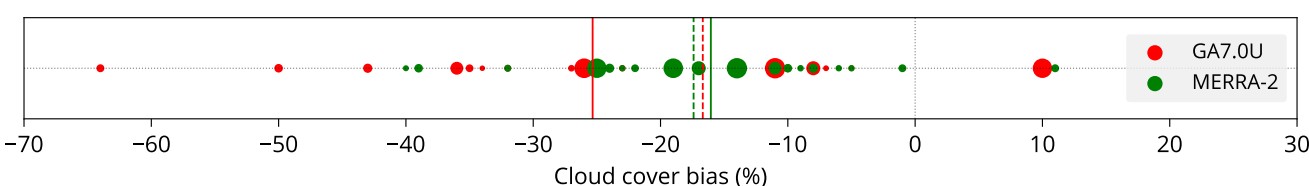

**Figure 6.** Cloud cover bias in models relative to observations. The points represent subsets as in Figure 5. The size of the circles is proportional to the number of days of observations in the subset. The solid lines are averages, and dashed lines are averages weighted by the number of days.



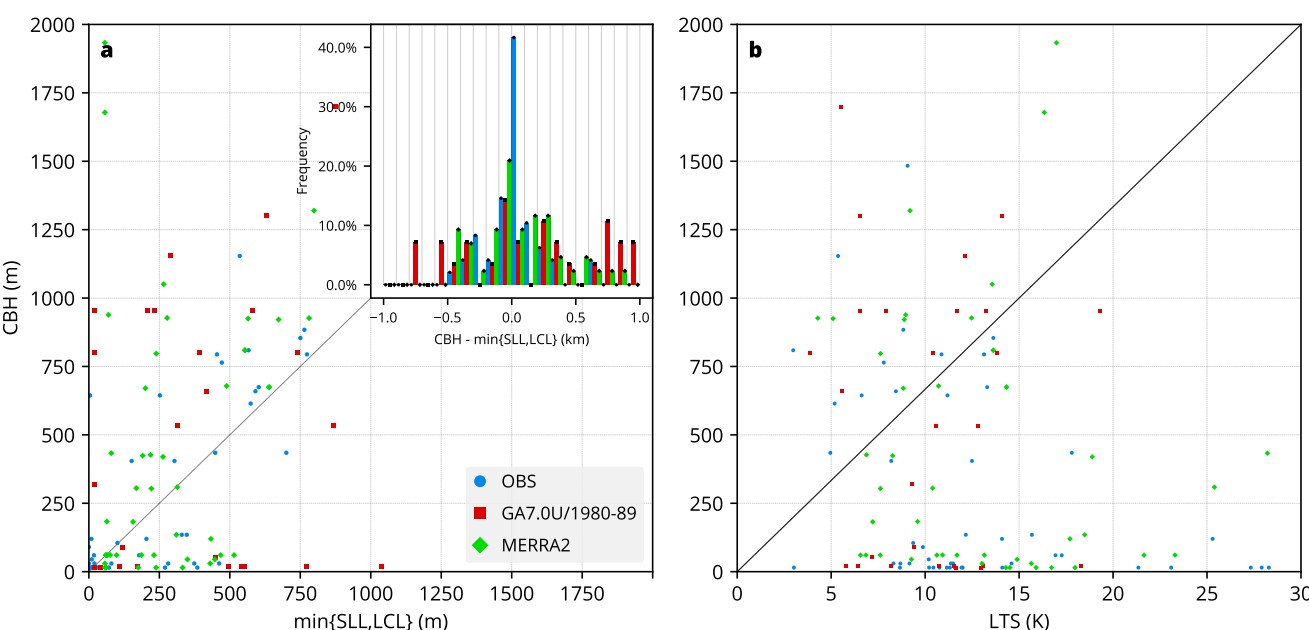

**Figure 7.** Scatter plots of radiosonde measurements on the TAN1802 and NBP1704 voyages between December and May (inclusive) and 60–70°S latitude. Corresponding profiles from GA7.0U/1980-89 and MERRA-2 are selected, i.e. having the same geographical coordinates and the same time of the year. Each point on the scatter plots represents a radiosonde measurement. The plots compare three datasets: observations (OBS), GA7.0U/1980-89 and MERRA-2. The points of GA7.0U/1980-89 (free-running) are selected randomly from years 1980 to 1989 of the simulation. The radiosonde measurements are matched with ceilometer (OBS) and COSP-based CBH (GA7.0U/1980-89 and MERRA-2). **(a)** shows the points as a function of min{SLL, LCL} and CBH. The inset histogram shows distribution of the difference of CBH and min{SLL, LCL} in bins of 100 m, where each bin contains three bars for the three datasets. **(b)** shows the points as a function of LTS and CBH.





**Figure 8.** SLL distribution panel plot. The plots show histograms of SSL as a function of pressure derived from radiosonde measurements (OBS) and model fields (GA7.0U, MERRA-2) **(a–f, m–r)** and scatter plots of cloud base height (CBH) vs. minimum of SLL and LCL corresponding to the plots above **(g–l, s–x)**. The numbers at the top of each panel indicate the number of soundings which make up the histogram and the percentage of sea ice cases as determined by a NSIDC satellite derived sea ice concentration product. The histograms and scatter plots are binned by season and latitude (column) and voyage (row).



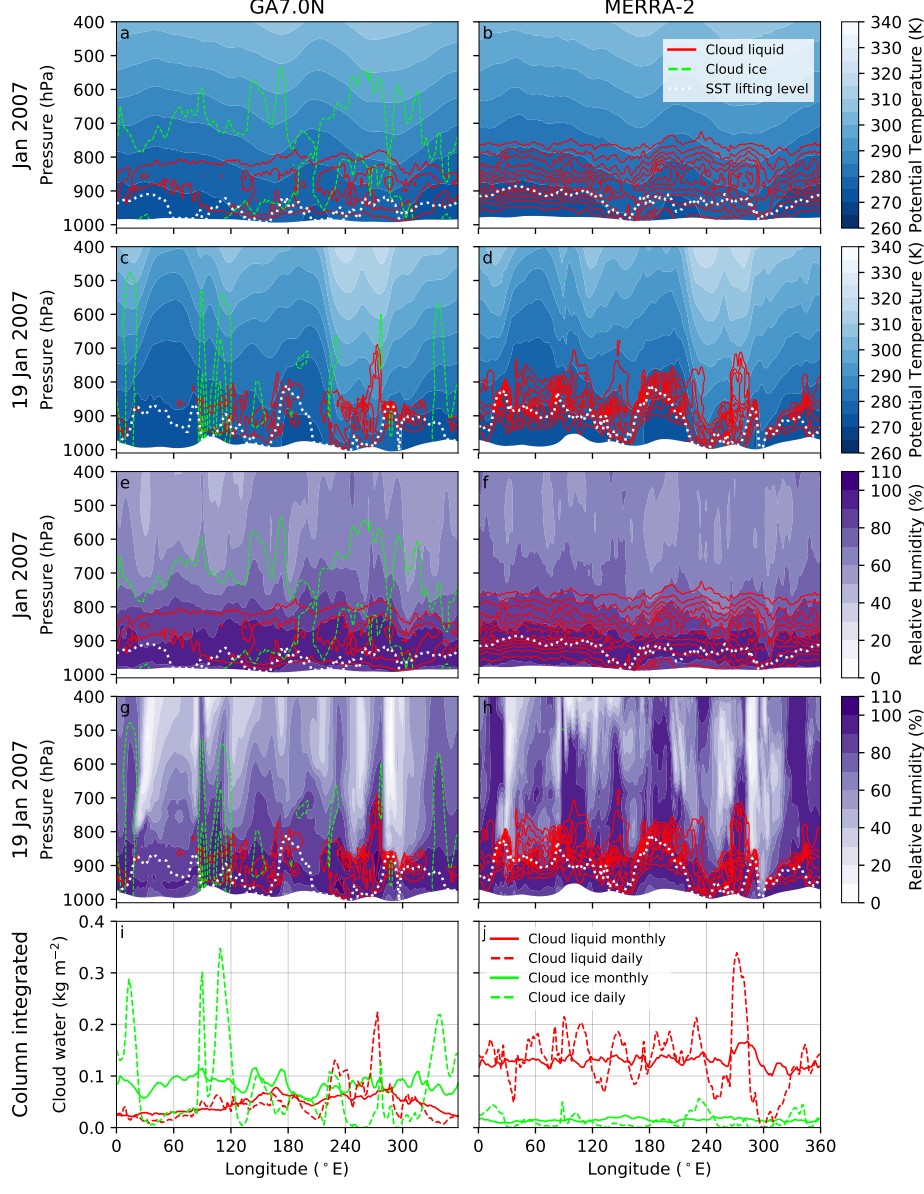

**Figure 9.** Zonal plane plot of cloud liquid and ice mixing ratios in GA7.0N and MERRA-2 at 60°S. The cloud liquid and ice mixing ratios are plotted as contours on top of the potential temperature fields **(a–d)** and relative humidity fields **(e–h)**. SLL is indicated by a white line. **(a)**, **(b)**, **(e)**, **(f)** show a monthly average in January 2007 and **(c)**, **(d)**, **(g)**, **(h)** show a daily average on 19 January 2007. **(i)**, **(j)** show the column-integrated values of cloud liquid and ice water as a function of longitude corresponding to the plots above, January 2007 ("monthly") and 19 January 2007 ("daily").





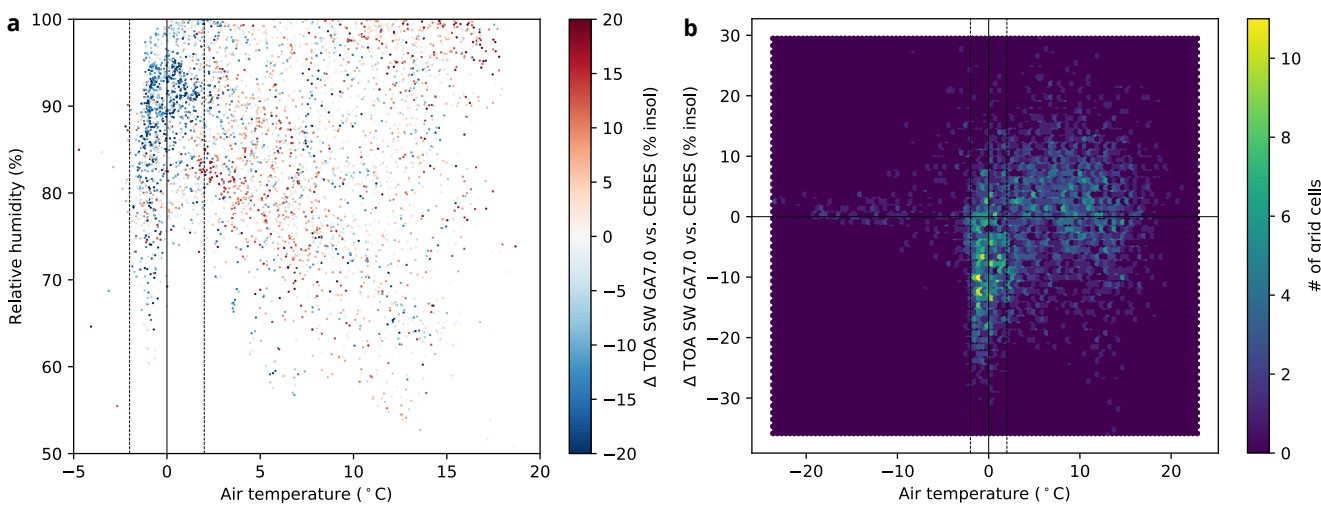

**Figure 10.** Scatter **(a)** and density **(b)** plot of SW radiation bias in GA7.0N/2007 grid cells between 40°S and 70°S as a daily average on 19 January 2007. **(a)** shows SW radiation bias as a function of near-surface air temperature and near-surface relative humidity. **(b)** shows the density of points as a function of near-surface air temperature and the SW radiation bias. The bias is expressed as a percentage of the incoming solar radiation in the grid cell. Each point represents a single model grid cell. -2 and +2 °C air temperature is marked by dashed lines.



| Voyage | Ship | Org. | Start | End | Days | Region | Lat. | Lon. |
|---|---|---|---|---|---|---|---|---|
| TAN1502 | RV *Tangaroa* | NIWA | 2015-01-20 | 2015-03-12 | 51 | Ross Sea | 41°S–75°S | 162°E–174°W |
| TAN1802 | RV *Tangaroa* | NIWA | 2018-02-08 | 2018-03-21 | 41 | Ross Sea | 41°S–74°S | 170°E–175°W |
| HMNZSW16 | HMNZS *Wellington* | RNZN | 2016-11-20 | 2016-12-20 | 20 | Ross Sea | 36°S–68°S | 166°E–180°E |
| NBP1704 | RV *Nathaniel B. Palmer* | NSF | 2017-04-11 | 2017-06-13 | 63 | Ross Sea | 53°S–78°S | 163°E–174°W |
| AA15 (AA V1–V3) | *Aurora Australis* | AAD | 2015-10-22 | 2016-02-22 | 123 | Indian O. sector | 42°S–69°S | 62°E–160°E |

**Table 1.** Table of voyages. The table lists voyages analysed in this study. Listed is the voyage name (Voyage), which is the official name of the voyage or an abbreviation for the purpose of this study, ship name (Ship), organisation (Org.), start and end dates of the voyage (Start, End), number of days spent at sea (Days), target region of the SO (Region), maximum and minimum geographical coordinates of the voyage track (Lat., Lon.).





| Instrument/Voyage | AA15 | TAN1502 | HMNZSW16 | NBP1704 | TAN1802 |
|---|---|---|---|---|---|
| Lufft CHM 15k | | | ✓ | ✓ | ✓ |
| Vaisala CL51 | ✓ | ✓ | | | |
| iMet radiosondes | | | | | ✓ |
| Radiosondes (other) | | | | ✓ | |

**Table 2.** Table of deployments. The table cells indicate if data from a given instrument (row) was available from a voyage (column).