# Peer review of "Evaluation of Southern Ocean cloud in the HadGEM3 general circulation model and MERRA-2 reanalysis using ship-based observations"

_Atmospheric Chemistry and Physics, 2019_

## Referee Comment (RC1) · Anonymous Referee #1 · 8 May 2019

Review of "Evaluation of Southern Ocean cloud in the HadGEM3 general circulation model and MERRA-2 reanalysis using ship-based observations" by Kuma et al. (acp-2019-201)

Summary:

The paper investigates cloud cover over the Southern Ocean through comparisons between numerous ship-based measurements and model outputs (including reanalysis). They demonstrate underestimation of low-level cloud cover in the HadGEM3 model and in the MERRA2 reanalysis. They investigate the link between boundary layer thermodynamics and low-level cloud cover and cloud biases. They show that the

[Figure]

TOA SW biases are mainly related to places where the coldest near-surface airmasses are (near or below zero). They conclude on the subgrid-scale parameterisations being responsible for misrepresentation of clouds in model rather than boundary layer thermodynamics.

Relevance of the paper and overall comment:

The paper presents and describe a very valuable dataset of ship-based measurements of low-level cloud over the Southern Ocean, where observations are badly needed to understand the near-surface processes affecting cloud formation and responsible for the cloud/radiative biases in climate models over the SO. To this respect the paper addresses relevant science questions in the scope of ACP. However, it seems to me that more work is needed to achieve ACP standards, in the way the science is presented and discussed (major revision). The dataset deserves better scientific discussion and less vague or speculative comments in several parts of the paper. Figure 5, 7,8 and 10 are very interesting but the analysis and discussion should be better handled. I first list some major comments, and then line by line comments.

———————————————- Major comments: ———————————————-

1)

The use of different time-periods needs to be much better introduced, justified, and discussed. I don't understand why the author use GCM simulations for the 1980-1989 period in a free-running mode, and then a nudged simulation for the year 2007 (only), while MERRA is used only for the 2015-2018 (the years where ship-based measurements took place). The reader needs much better justification for the choice the authors make to compare different periods. And a discussion on the shortcomings of doing so should appear in the paper. P4-Line 26, the authors say "Limited data availability meant that no nudged runs were available for the period 2015-2018". Is this really the case? And if this is the case, why not having a free-running simulation for this period then? And why is the nudged run over 2007 only? Also, MERRA2 could

be used for the 1980 period. MERRA2 is available for >=1980. MERRA could help bridge between the period 1980's/2007's of the GCM outputs and the period of the ship-based measurements (2015-2018). At least using MERRA2 also for the 1980's and 2007 + explaining/discussing the choice for the time periods of the GCM runs would be needed. The best case would be to have GCM runs over 2015-2018. How using different periods for GCM/MERRA2 would affect Figure 5 for instance? And what about Figure 1 and the TOA biases where only the year 2007 is shown? The authors speak about the years 2016-2018 that had unusually low sea ice extent (p15-Line16): how does this impact the comparisons with other years where sea ice was different? Having said that it is possible that the paper could be improved by giving up Figure 1 or 2, while focusing more on the novelty of this work, which is the ship-based measurements of clouds (+thermodynamics), and drop the comparisons to GCMs (mainly because they are different periods...) and keep the comparisons to MERRA2, and add the ERAI reanalysis. To this respect ERAI, which is widely used, could also be presented here. How is ERAI doing vs. MERRA2. Say, if ERAI has a contrasting behavior to MERRA2 (ie more like the GCMs) the authors could consider presenting the ship-based measurements + MERRA2 + ERAI over similar periods, and drop the GCM runs, which deal with other periods.

2)

The discussion of ship-based measurements should be better related to the TOA SW bias over all periods where these field measurements were available. Since the authors try to understand what the models are doing wrong, the discussion should make the most of the different measurements period. Because maximum insolation occurs in January, a focus is made on this month but Figure 2 clearly shows that March – for instance – can also and still show substantial biases. Since Ship-based measurements are also available in autumn and November-December, it would be very welcome to have also biases like the ones shown in Fig. 2 for the autumn and other summer months. Is the TOA SW bias spatial pattern (and the related comparisons between

models) the same during these other months? I suggest Figure 3 to show only biases (the subplots m-p) for summer and autumn. The other maps (a-l) are difficult to read with the blue-shaded colourscale and I am not convinced they need to remain. Better discussing the cloud cover results in Summer/Fall (Figure 5) in relation to radiation biases in Summer/Fall would improve the overall discussion and conclusions of the paper. This would allow to make the most of the ship-based observations. Figure 5 is a great one and it would deserve better discussion in light of the motivations (i.e. the TOA SW biases over the SO and why theses biases are present). The authors note that GA7.1 reduces the SO SW radiation bias (e.g. in the abstract p1-L9). Figure 5 does not show any cloud from GA7.1 (only GA7.0). Why GA7.1 is performing better? Is this really because of better cloud representation (but we cannot see it from Figure 5)? And if not, what does it say about cloud being the main/only reason of SW radiation bias? Related to 1), what ERAI would give in Figure 3? More like the GCMs or like MERRA2? Perhaps ERAI brings this contrasting behavior that the authors highlight between GCMs and MERRA2, and this would allow to have both observations and reanalyses (only) used over the same period (2015-2018)

3)

The authors tend too often to rely on previous conclusions from previous papers (e.g. the Bodas-Salcedo et al. ones) to comment on what they find, rather than more thoroughly commenting/discussing their own novel results. The discussion part for instance gives much room to results of previous published study and/or to speculative comments about why GA7.1 is doing better than GA7.0 etc. and how MERRA2 is overcompensating for the low cloud-cover etc. Several sentences using "we cannot conclude...", "we cannot make the same conclusion... but it seems plausible..." "we cannot make substantial conclusions..." considerably weaken the discussion (section 4) from the beginning, and hence the paper, while it seems that all the ship-based measurements bring very valuable results (Figure 5, Figure 7, Figure 8) and interesting comparisons to MERRA2, and GCM runs (but cf. my point 1. on the time periods used).

4)

The discussion on the effect of sea ice is overlooked while it seems that some discussion could be made from Figure 8 (q,r and w,x). Also, while it seems that 8w is still showing some correlation, Figure 8x shows very different behavior and no attempt is made to comment on this. Given that a lot of the soundings you use (65) were made in 100% sea ice regions, and many of your CBH observations as well (I see the number of points present in your Figure 8x compared to the other similar subplots), I would expect to see more in-depth study of these observations, and this is really missing the present version of the paper in my opinion. For instance, the recent study by Jolly et al. (2018) that you cite showed the influence of different regimes on cloud cover: can the observation in Figure 8x be explain by particular synoptic-scale regimes or just by the sea ice being 100%? And why? Also, that other recent study by Listowski et al. (2018) that you also cite showed that not all low-level clouds anticorrelate with sea ice fraction but only the liquid-bearing ones. Can the behaviour you see in Figure 8x also be explained by clouds being of different sort/phase? The absence of correlation between CBH and min(SLL and LCL) may lead to think that you could be observing clouds advected from other places not related to local atmospheric conditions, that may be different in nature/phase from clouds over open water (you mention some hints towards the detection of supercooled liquid water with one of your instrument, can't you improve the Figure 8 by adding information on the phase, notably for Figure 8x?) In other words, can what you are observing from regions with 100% sea ice be explained by changing synoptic scale regimes, or cloud phase, or other things? Speaking of the sea ice regions, in Figure 7 the very low CBH are identified as being due to fog/very low clouds (p13-L20) and these are the points we also see in Figure 8x. Could this be blowing snow since we are in a 100% sea ice-covered region where snow can accumulate? In relation to cloud phase, in Figure 9 you compare LWP and IWP for GA and MERRA only for a specific year and month (jan. 2007). This does not seem satisfying to conclude for the longer time scales/other periods. Here again the use of different time-periods in the paper is not very welcome (see my point 1.). Do you really need

Figure 9? If you really want to go into the cloud phase, using the lidar observations to assess the nature of cloud phase would be welcome. Or, as suggested in 1), perhaps only using MERRA2, ERAI (to contrast with MERRA2?) over 2015-2018 (only) would be a better option rather than using 2007, i.e almost a decade before the 2015-2018 ship-based observations...

5)

Finally, the authors say that subgrid-scale processes should likely be responsible for the cloud misrepresentation in models rather than the boundary layer thermodynamics but it is never said and commented on what these subgrid-scale processes are. Do you mean the microphysics? Other processes? A discussion of what is used in the models regarding these processes would perhaps help to understand what should be improved in priority in the models and why the models are wrong. Using the contrasting behaviors of models to try to pin down the cause of cloud misrepresentation is an interesting method but the authors should provide with more clues in the discussion about what those subgrid-scale processes are and try to spot the main differences in the way the models implement these processes.

———————————————- Line by line comments: ——————————————-

————————— Abstract ————————

P1-L9 By how much GA7.1 reduces the bias? P1-L17 The analysis you mention is referring to your Figure 9 and the related comments. They only refer to the period January 2007... as mentioned in my major comment 4) this is not satisfying I think. When one reads the abstract it seems that you compare modelling and MERRA2 over the same period as the ship-based measurements, which is not the case. This is misleading.

————————— 1.Introduction —————————

P3 – L12 : "It was also more..." : what does "it" refer to exactly ? P3 – L14 : "more

likely to have intermediate cloud fraction" This is not clear. What is meant here by "intermediate cloud fraction"? P3 – L19-20 Please double-check and be more precise here (what "tuning" do you mean?). Kay et al. (2016) changed the threshold temperature below which detrained condensates are ice crystals and not liquid any more. They lower this threshold, allowing for more condensates to remain in the supercooled liquid phase when being detrained. The way the sentence is written suggests that ice crystals only are detrained. P3 – L25 The reference to Jakob (2003) is a bit short or can be removed unless you specify what you mean by "cloud evaluation" regarding this specific study. P3 – L27-35 Please make a new paragraph and give section numbers to help the reader.

—————- 2. Method —————

General comment: I would suggest a section 2. Datasets and 3. Method (lidar simulator). As it stands, it seems that this section combines too many different information about the data/methods used in the paper.

P4 – L2 As mentioned in my major comments, adding ERAI would be very interesting since this is a widely used reanalysis by the community, and would allow to contrast MERRA2 on same time-periods than ship-based obs. P4 – L9 I wonder whether a small appendix summarizing the main aspects of the lidar simulator would not be needed here, since the reference put is a paper in prep.

P4 – L16 What is the difference between GA7.0 and GA7.1 ? This would help understand and discuss the better performance of the latter in terms of SW bias (as stated in the abstract).

P4 – L 18 As said in the major comment it should be explained why these runs are used. 1980-1989, and then 2007. Why not having runs over more recent periods (as the ship-based measurements).

P4 24 – "Can only be compared statistically" What do you mean? Please clarify.

[Figure]

P4 – L26 "Limited data availability..." What do you mean? See my major comment 1)

It does not seem that you are saying over which period you analyse MERRA2. This should appear in this section.

P5 – L15 "downsampled" from what initial resolution?

P6 – L5 "appears largely zonally symmetric" I don't think we can say the pattern of the bias is symmetric, even zonally, but rather that the bias is present across all longitudes, but its magnitude does change zonally.

P6 – L6 "with a notable exception..." Precisions not needed in the section presenting the ship measurements...

P6 – L8 "Figure 1..." This Figure is already mentioned before P5 – L21.

P6 – L20 I am not sure what is meant by "directly reveal the cloud liquid...". The strength of using a simulator is to compare the observables and not to rely on all the hypotheses used by inversion routines to retrieve IWC and LWC from lidar observations.

P7 L24 – Please clarify the title e.g. "Geographical areas/domains investigated" or "Domains used for the analysis"

Also, having 2.1 as "Datasets" then 2.2 as "Domains" then 2.3 "COSP simulator" is not ideal I think, and I would first present all datasets and tools, and then the domains.

P8 – L12 The title of this subsection is misleading since you are not using COSP simulator in the end, but your own simulator. Please change the title accordingly.

It seems to me you don't need a section 2.3 and you could have everything put in current 2.4.3 where you could at once explain the modeling of the lidar signal along with its processing.

P9 L15 What is this known value of LR? Where does it come from?

P9 L21-22 Citing Kotthaus et al. at the end of the paragraph falls a bit short and I am wondering if it should not appear earlier in the paragraph with some more explanation about why you refer to this study. Are you using their method? Then please say it.

P10 L7-9 Why do you need to do this? How are these random samples used then?

To shorten this section 2. I would not define SLL here, rather when it is used for the first time. Also SSL is neither a dataset, nor a tool, rather a variable defined to help with the analysis.

Also, is there any past reference using this definition? If yes, please cite relevant paper.

————————————- 3. Results ————————————-

P10-L26 to P11 L-16 There are too many statements dealing with observations made on Figure 3 that are actually difficult to see, whereas Figure 4, introduced after, is more helpful to confirm statements made by the authors. Also, as suggested in major comments, I would tend to simplify Figure 3 by showing only the biases and remove all the blue-shaded figures where the biases are difficult to read, especially regarding the statements made by the authors in the main text.

P10 – L28 "Lower" than what? And what biases? Please clarify.

P10 – L29-30 I would remove the sentence about the "predominantly zonally symmetric pattern" and the "more variable patterns in the tropics", which is not very clear to me.

P11 – L1 "upwelling and downwelling" what?

P11 – L3 "large differences" between what? I would drop the mentions to the Peninsula and what is happening to the east of it as it is not clear why one would give so much importance to this since the ships did not get there anyway.

P11 – L1 I don't understand the footnote. Also, I am not convinced there is a need to highlight a particular day in the present paper.

P11 – L4 One cannot really see this "greater reflectivity".

P11 – L13 "With some individual cloud systems being too bright". I am not sure this should remain in the text. Again, I think all the consideration about the blue-shaded maps in Figure 3 (but biases maps should be kept) should be removed and Figure 4 should be used instead.

P11 – L21 "cyclical". Rather "seasonal"?

P11 – L20 What is meant by "likely a secondary modulating factor". Please be more explicit. A modulating factor for what?

P11- L26 "These panels also justify why..." Not needed.

P12 L4-6: The two sentences fall a bit short. Also, they would be in better place in the discussion part, with more explanations. "....in the GA7.1 model": so what?

P12 L2 Figure 5 is very interesting and rich, and more analysis should be provided also regarding similarities or differences between summer results and autumns results. (Please consider adding letter to designate specific subplots of Figure 5). Also it seems that obs and model agree more where the statistics is larger (more days), can't you say something about that? Isn't it possible that at other time/places the larger disagreement between model/obs is partly due to smaller statistics of observations? This relates to my major comments that more analysis and discussion are really needed on this plot.

P12-L10 What period is used for MERRA2 here?

P12 L12. As mentioned in the major comment. Why can the authors trust comparisons between simulation of the 1980s period and the 2015-2018. This should be much better introduced/justified.

P12 L19-20 how much higher?

P12 L20-21 "Due to the zonal...of the whole SO" Could suit the discussion part. Not needed here.

P12 L27-34 I would drop Figure 6 and give only numbers. It saves a Figure. Also what bothers me is that GA7.1 is said to be better from nudged simulations but, in the end, only GA7.0 is presented here, because of the decadal run being only available with GA7.0. This is again a shortcoming of accepting to work with so many different time periods for different simulations.

P13 – L1-5 Have the authors consider to use satellite data, or to rely on previous publications to try to assess how the comparison to models is biased by extinction of the ceilometer signal into the lowest thick clouds? At least this should be discussed in the discussion part. This is not the case now.

P13 L10-11 Is the extraction made above the lat/lon of the balloon launch or does it follow the radiosonde trajectory? I guess it is the latter but you may want to clarify this in the text.

P13L11 Can you make a subplot for each of the dataset? It is difficult like this to spot differing behaviours between coloured markers.

P13 – L14 What relationship?

P13 – L19 How large?

P13- L23 how weaker?

P13 – L25-27 The fact that LTS is not a good indicator should be discussed in the discussion part and I don't think it is the case for now. This relates to my major comment 3) where I suggest that more emphasis should be given in the discussion to all results obtained from these novel ship-based measurements.

P13 – L28-34. Figure 8 is introduced, but then some general statement are made about synoptic scale forcing. It would be much better, for the reader, to stick to the Figure.

P14 – L1 "As can be seen…where there is no sea ice". What can be said about Figure8a and b where there is no cloud but at the same time GA7.0 is not in agreement

at all with observations? Also what is the unit in the x-axis of subplots Figure 8a-f and Figure 8m-r?

In Figure 8g-l and s-x, you are not showing the modelled dots, only observations. I would have expected to see the model outputs as well. Or is it not useful here?

P14L3-4 "There is no substantial difference between..." This is not true for Figure 8a and b... which present non-sea ice cases. This should be discussed. "Plausible effect"? What do you mean?

P14 L8 – What is meant by subgrid-scale processes? Please be more specific.

P14 - L2 I am not sure about this subsection. I struggle with having it only focusing on January 2007. Since the novelty of the paper is the ship-based measurements I am not sure having this part here is relevant, especially that it is only about comparing Jan 2007 for two models. Plus, the GA7.0 one is not the one used in Figure 5, but the nudged one, and it is not clear what period is used for MERRA2. Why not showing also GA7.1 since it is spotted as reducing the SW biases (because of the modelling of larger supercooled LWP?)

P14L26-30 "We should note..." These are comments for the discussion part, but even so, these considerations are also and already mentioned in other places of the paper and remain very general and a bit speculative. I am not sure these zonal plots deserve a separate section, also because of these time period issues mentioned above.

————————————— 4. Discussion —————————————-

In general the discussion should be more focused on your results at least in the beginning and spend less time on explaining previous works. Figure 10 (which is interesting indeed) should come earlier in the discussion. Also, you don't seem to do discuss Figure 10b, but only Figure 10a.

Sentences like (P15-L18-21) "Combined..." are a bit speculative and more room should be rather given to discussing the results obtained from ship-based measurements, ie. Figure 5, Figure 7 and Figure 8, and 10. And then make the link to the TOA SW bias issue and relate it, possibly, to the LWP as modelled (cf. Figure 9 – if still considered relevant in a revised version).

P15-L34 to P16 L4. This is too much about other study, not enough discussing your results. Figure 10 comes after that and this is not appropriate. Also – as an example of additional discussion element – are the ship-based observations, which show larger discrepancies from MERRA2, in places where the near-surface temperature is the coldest? In other words, can you relate Figure 10 with your cloud results, instead of only speaking of the SW bias?Also, why is Figure 10 only showing the year 2007? Why not showing the decadal simulation, and the MERRA2 outputs as well (during the ship-based measurements)? What do they say? How is it consistent or not with the cloud simulations in these models? These sorts of analysis/discussions are really missing in the paper, in my opinion.

P17 L9 "Because sea ice is an important factor. . .": What is meant by "secondary effect on cloud cover"? It seems to me you have the opportunity to say something about the effect of sea ice on very low clouds (and specifically the ones missed by satellites) – e.g. your Figure 8x – but you are not exploring this in the paper. This goes along with my major comments that not enough efforts are made to discuss the very interesting observations you have from ship over three years and in sea-ice free/covered regions.

——————————— 5. Conclusion ———————————

In the conclusion only you speak again about the sugrid-scale processes without specifying them. This should be a paragraph on its own in the discussion part, trying at least to understand how the various models are doing different in parameterising these processes. This would give more perspective to the present work I think.

——————————- Figures. ———————————-

Figure 2 If you still want to keep all the model results (provided you better justify your

method – see my major comments) then you should add the time-periods for the simulations you use, and for the observations, so that one immediately knows you are using different times for comparisons (and that this is then discussed in the text).

Figure 3 As I said before, one struggles to see features with a single colour-shaded scale. As suggested I would keep only the plots showing the biases, and for summer and autumn (as these are seasons investigated with ship measurements).

Figure 4 The horizontal line indicates the "0" value for the bias (red curve). Please make it red (and thicker, or dashed).

Figure 5 What period is used for MERRA2? Why not also showing the nudged runs with the better (according to what you say) version GA7.1U.

Figure 6 Not sure this figure is needed. See my comment in the relevant section.

Figure 7 This would be better to separate the dataset in different subplots to see the different behaviours.

Figure 8 What are the x-axis units in the subplots a-f and m-r? The markers in the g-l and s-x subplots are quite small. Can you either make them larger or increase the size of the subplots.

Figure9 What are the contour values?

---

## Referee Comment (RC2) · Anonymous Referee #2 · 16 Jun 2019

Review Kuma et al: ' Evaluation of Southern Ocean cloud in the HadGEM3 general circulation model and MERRA-2 reanalysis using ship-based observations' ( MS No.: acp-2019-201) The authors conducted analysis of three model datasets by focusing on the Southern Ocean to understand errors in models in the shortwave (SW) radiative flux at the top-of-the-atmosphere, using ship observational dataset as well as satellite observations to understand the errors. They found that GA7 runs and MERRA-2 runs have the opposite bias in the outgoing SW flux (underestimate in GA7, overestimate in MERRA-2) over the southward latitude of 55S. They compared their cloud amounts with the ship observations and showed that both models underestimate their cloud amounts. They also conducted nudged-runs and showed that there is a big dif-

ference in cloud liquid water amount in these models, concluded that the main source of the difference in their SW bias is from the difference in their cloud properties, which are determined by the sub-grid cloud parameterizations. The shortwave bias over the Southern Ocean tends to be a common problem in climate models. This is a nice piece of work which contributes to improve our understanding of the representations of clouds over the region. However, current manuscript misses some information for their logic to convince readers, hence the key message remains unclear. I suggest this paper to be published after a minor revision. Main comments: Although GA7 runs and MERRA-2 runs have the opposite bias in the outgoing SW flux over the southward latitude of 55S, both HadGEM3 GA7 and MERRA 2 underestimate cloud amount. In Discussion section, the authors mentioned that models may fail to represent fog or low cloud which are generated by convection which are induced from subzero airmass from polar regions over warm water. What our community is keen to know is whether we can improve the representations of such clouds in GCM or we should seriously start thinking of using cloud resolving model or GCM. Whether/how much the underestimate of the cloud amount improves in their nudged runs will provide a clue for it. The authors should add a figure which shows cloud amounts in free run and nudged runs. The authors showed that main difference in SW radiative flux bias over the Southern Ocean between HadGEM3 GA7 runs and MERRA 2 runs is cloud water amount. This shows a big impact of subgrid cloud parameterizations on radiation. Please check subgrid cloud parameterizations in GA7 and MERRA2 then discuss which parameterization could potentially cause the difference in radiative flux. Since the authors showed the opposing sign of the SW CRE south and north of 55S in GA7.1, it would be useful to apply the same analysis (comparison to the ship observations, analysis of the nudged runs) to the region of the north of 55S, confirm whether the smaller error is because of the (less worse) representations of the cloud amount over the region. Minor comments: Discussion: the beginning (L1-10) was difficult to read, because the authors mention the opposing sign of the SW CRE south and north of 55S in GA7.1, but then solely talk about the results over the south of 55S. Figure 6: Clarify what is the weight for the

weighted average. Figure 8: add grid values to the Frequency axis P11-l1: 'upwelling and downwelling' Where are regions of upwelling and downwelling radiative flux? If the authors are talking about large scale circulation, these should be 'ascent and descent'. P11-l4: I cannot see the results described about models. And the contrast between western and eastern sides of the Antarctic Peninsula contradicts to the following description 'The zonal symmetry. . ..' P11-l14: Figure 3p? P11-l32: 'consistently positive': negative in Sep-Dec in 60S-70S P11-l33: 'also lower than GA7.0 and GA7.1': not necessarily in GA7.1 P13-l14: Did you define SLL and LCL? (Super liquid level and lifting condensation level?) How did you define SLL? P13-l22: Give a speculation why min(SLL, LCL) is better correlated with CBH than SLL/LCL individually. P14-l5: Provide a figure or reference about SLL in GA7.0 is higher than observed. P14-l16: Fig 9. It is not clear why the authors create these plots over two different backgrounds. P14-l18: Fig 9. Not clear. Different colors should be used for different levels to show this. P14-l29: cloud cover a reduce ..': typo? Fig 5: Why did the authors exclude 50S-55S for the plots? Fig 8: The authors did not analyze model results in other latitudes where clouds shows the opposite bias (in 50S-55S). P15-l10-11: I cannot follow the logic here. P16 l23: Is it possible to add the definitions of supercooled liquid in GA7.0 and MERRA-2? P17 l11: Is this a result from the nudged run or from other studies?

---

## Author Comment (AC1) · 21 Sep 2019

The response to referees is contained in the supplement.

Please also note the supplement to this comment:
https://www.atmos-chem-phys-discuss.net/acp-2019-201/acp-2019-201-AC1-supplement.pdf

---

## Referee Report (RR1)

**Review of revised version of "Evaluation of Southern Ocean cloud in the HadGEM3 general circulation model and MERRA-2 reanalysis using ship-based observations" by Kuma et al.**
**(acp-2019-201)**
* * *
**Main comment:**
* * *
I am satisfied with the improvements made by the authors to the manuscript, who made the efforts to address all my concerns. They have notably addressed the main issue which were the different time ranges used in the various simulations analysed. They better discuss their results. There are still few statements regarding what the authors try to demonstrate, which are not clear to me, esp. regarding GA vs MERRA performances. This should be improved before publication by either toning down some statements, or better demonstrating them. (minor revision)
* * *
**Line by line comments (using the latexdiff document):**
* * *
P3-L14: You should give a reference for AMPS:
Powers, J. G., Manning, K. W., Bromwich, D. H., Cassano, J.J., and Cayette, A. M.: A decade of Antarctic science sup-port through AMPS, B. Am. Meteorol. Soc., 93, 1699–1712,https://doi.org/10.1175/BAMS-D-11-00186.1, 2012.

P3-L22: the most recent study investigating SLW in the SO is the one by Listowski et al. 2019 I think, and it should appear here after Jolly et al. 2018:
Listowski, C., Delanoë, J., Kirchgaessner, A., Lachlan-Cope, T., and King, J.: Antarctic clouds, supercooled liquid water and mixed phase, investigated with DARDAR: geographical and seasonal variations, Atmos. Chem. Phys., 19, 6771–6808, https://doi.org/10.5194/acp-19-6771-2019, 2019.

P6-L13: AA15 is not labelled in Figure 1. I guess it is AA V1-V3.

P6-L14-15: Please double-check the sentence:
→ "…is present at all longitudes in the SO (section 5.1), affected by atmospheric circulation in the SO (…). ("in the SO" repeated twice)

P7-L9: Define here GNSS (this is only done at L18 for now).

P15-L27:  peaking below 500m (not "km")

P18-L2: there is no label b4, do you mean c4?

P18-L2-L3: This is not so clear or please explicit why you can say this. As it can possibly be said for (a1) and (a2), this is not so sure about the other sea-ice free cases.

Please explain (by quantifying biases?) how you see in Figure 8 that MERRA-2 is worse than GA7.1N.

P18-L34: I am not sure to agree ("is majority liquid"). Looking at Figure 9i: IWP>LWP!

P19-L3: is almost entirely liquid. Not ice (looking at Figure 9j)

P19-L9: in both models

P19-L14: similar

P20-L14: "Remarkably, the observed and simulated cloud occurrence profiles do not appear to be significantly different between the DJF and MAM seasons or different latitude bands between 55 and 70°S (Figure 5)": I am not sure to agree with the authors, unless you specify what you mean by "significantly". Looking at all the profiles in Figure 5 there are clear differences (multiple layers, single low layer, etc.). So, I am not sure about the rest of the paragraph either ("This is in contrats…"). Please clarify and improve this part.

P21-L6-7: Following my comment on P18-L2-L3 I am not sure to agree with this or this is not convincingly demonstrated.

P21-L29: Are you referring to cold air outbreaks here? They tend to form cloud streets, and one expects models to struggle with these low clouds. Any reference here to back up your comment?

P21-L35: Rewrite here the references used in the intro, after "…in summer months"

P22-L31: Please give at least one reference here.

P23-L15-30: Since you define a new quantity in your study (SLL), this would be interesting to recall it here, and explain the benefit of using it.

**Figures**:

Figure 9: what are the contour labels? On subplots c, d, g, and h, one cannot see at all the different red contours. Please improve the figures, and indicate contour values (in the caption or in the plot).

---

## Author Response (AR2)

**Response to referees on *"Evaluation of Southern Ocean cloud in the HadGEM3 general circulation model and MERRA-2 reanalysis using ship-based observations"* by Peter Kuma et al.**

Peter Kuma[1], Adrian J. McDonald[1], Olaf Morgenstern[2], Simon P. Alexander[3], John J. Cassano[4], Sally Garrett[5], Jamie Halla[5], Sean Hartery[1], Mike J. Harvey[2], Simon Parsons[1], Graeme Plank[1], Vidya Varma[2], Jonny Williams[2]

[1]School of Physical and Chemical Sciences, University of Canterbury, Christchurch, New Zealand
[2]National Institute of Water and Atmospheric Research, Wellington, New Zealand
[3]Australian Antarctic Division, Kingston, Australia
[4]Cooperative Institute for Research in Environmental Sciences and Department of Atmospheric and Oceanic Sciences, University of Colorado, Boulder, Colorado, US
[5]New Zealand Defence Force, Wellington, New Zealand

We would like thank the referees for their valuable comments. We have addressed a number of related referee comments by replacing the free-running model GA7.0U/1980-90 with a nudged model run GA7.1N/2015-2018, nudged to observations based on the observatioanl HadISST SST and sea ice dataset and the ERA-Interim reanalysis. The new model performed much better in terms of cloud representation relative to observations, but a significant error in TOA outgoing SW radiation and cloud occurrence representation remains, especially related to low cloud and fog. This is reflected in the revised manuscript.

The referees' comments below are marked in **bold**, followed by authors' response. We supply a latexdiff document which identifies the changes made. Page and line numbers in our response comments refer to the latexdiff document.

Introduction to Figure 10 in the text has been relocated to Results, but for clarity we keep the original numbering of figures, which can be changed in a final revision.

**Anonymous Referee #1**

**Review of "Evaluation of Southern Ocean cloud in the HadGEM3 general circulation model and MERRA-2 reanalysis using ship-based observations" by Kuma et al. (acp-2019-201)**

**Summary:**
**The paper investigates cloud cover over the Southern Ocean through comparisons between numerous ship-based measurements and model outputs (including reanalysis). They demonstrate underestimation of low-level cloud cover in the HadGEM3 model and in the MERRA2 reanalysis. They investigate the link between boundary layer thermodynamics and low-level cloud cover and cloud biases. They show that the TOA SW biases are mainly related to places where the coldest near-surface airmasses are (near or below zero). They conclude on the subgrid-scale parameterisations being responsible for misrepresentation of clouds in model rather than boundary layer thermodynamics.**

**Relevance of the paper and overall comment:**
**The paper presents and describe a very valuable dataset of ship-based measurements of low-level cloud over the Southern Ocean, where observations are badly needed to understand the near-surface processes affecting cloud formation and responsible for the cloud/radiative biases in climate models over the SO. To this respect the paper addresses relevant science questions in the scope of ACP. However, it seems to me that more work is needed to achieve ACP standards, in the way the science is presented and discussed (major revision). The dataset deserves better scientific discussion and less vague or speculative comments in several parts of the paper. Figure 5, 7,8 and 10 are very interesting but model analysis and discussion should be better handled. I first list some major comments, and then line by line comments.**

[Figure]

**————————————- Major comments: ————————————-**

**1)**

**The use of different time-periods needs to be much better introduced, justified, and discussed. I don't understand why the author use GCM simulations for the 1980-1989 period in a free-running mode, and then a nudged simulation for the year 2007 (only), while MERRA is used only for the 2015-2018 (the years where ship-based measurements took place). The reader needs much better justification for the choice the authors make to compare different periods. And a discussion on the shortcomings of doing so should appear in the paper. P4-Line 26, the authors say "Limited data availability meant that no nudged runs were available for the period 2015-2018". Is this really the case? And if this is the case, why not having a free-running simulation for this period then? And why is the nudged run over 2007 only? Also, MERRA2 could be used for the 1980 period. MERRA2 is available for >=1980. MERRA could help bridge between the period 1980's/2007's of the GCM outputs and the period of the ship-based measurements (2015-2018). At least using MERRA2 also for the 1980's and 2007 + explaining/discussing the choice for the time periods of the GCM runs would be needed. The best case would be to have GCM runs over 2015-2018. How using different periods for GCM/MERRA2 would affect Figure 5 for instance? And what about Figure 1 and the TOA biases where only the year 2007 is shown?**

The availability of model datasets was indeed limited by organisational capabilities. Especially, a nudged run for the observational period 2015–2018 was not available, therefore the choice of statistical comparison with a decadal simulation of 1980–1990. To address this major comment, we have produced a new nudged run of GA7.1 (UM 11.0) for the observational period. This nudged run was evaluated in the same way as MERRA-2 originally, i.e. in a 1:1 comparison, assuming that weather conditions are comparable between the model and observations. The results based on this run were more accurate than GA7.0U. We have decided to leave out the decadal run of GA7.0U entirely to improve clarity of the manuscript.

Due to this change, a significant part of the Results section (P13L11–P19L3) has been updated. Figure 5 shows significantly different results for GA7.1N (cloud occurrence bias is smaller), but the nature of the error has not changed – low cloud and fog is still underestimated in the model. Figure 6 shows that GA7.1N has lower average bias at 4–9%. Figure 7 shows that the correspondence between min{SLL,LCL} and CBH is still not well represented in the model. Figure 8 has been updated to show min{SLL,LCL} (previously it showed SLL), GA7.1N is matching the observed distribution quite well.

**The authors speak about the years 2016-2018 that had unusually low sea ice extent (p15-Line16): how does this impact the comparisons with other years where sea ice was different?**

The influence of low sea ice concentration in 2016–2018 may be hard to quantify with our data, considering that few of the ship observations are available prior to 2016. We therefore consider the whole ship-based dataset as representing relatively uniform conditions. The new nudged model run (as well as MERRA-2) are based on the observed sea ice concentration, therefore the 1:1 comparison of the models with observations should not be biased.

**Having said that it is possible that the paper could be improved by giving up Figure 1 or 2, while focusing more on the novelty of this work, which is the ship-based measurements of clouds (+thermodynamics), and drop the comparisons to GCMs (mainly because they are different periods. . .) and keep the comparisons to MERRA2, and add the ERAI reanalysis. To this respect ERAI, which is widely used, could also be presented here. How is ERAI doing vs. MERRA2. Say, if ERAI has a contrasting behavior to MERRA2 (ie more like the GCMs) the authors could consider presenting the ship-based measurements + MERRA2 + ERAI over similar periods, and drop the GCM runs, which deal with other periods.**

We did not include ERA-Interim in the new revision. Technically, ERA-Interim does not provide the necessary model fields for running the simulator. However, it would be possible to compare with ERA5, which provides these fields. The nudged run of GA7.1 is nudged to ERA-Interim. Therefore, the dynamical conditions are likely very similar, even though they can still differ in their representation of clouds. We did

not want to leave out the GA model because it is a model which the authors want to improve (the authors participate on development of this model).

We agree that comparison with another reanalysis such as ERA5 would be interesting, but we prefer to limit the scope to just GA7.1 and MERRA-2.

**2)**
**The discussion of ship-based measurements should be better related to the TOA SW bias over all periods where these field measurements were available. Since the authors try to understand what the models are doing wrong, the discussion should make the most of the different measurements period.**

In the updated manuscript we link Figure 5 with the SW radiation bias by introducing a "back-of-the-envelope" calculation showing how the SW radiation bias would change by increasing cloud cover in GA7.1N by 5%, assuming no change in cloud albedo (Table 3; P23L9-14). The error would be reduced by over a half by increasing the cloud cover alone by an amount which is approximately consistent with the results shown in the updated Figure 5 and 6.

**Because maximum insolation occurs in January, a focus is made on this month but Figure 2 clearly shows that March – for instance – can also and still show substantial biases.**

The updated Figure 2 now shows the biases. The bias in MAM (Figure 2h, i) has a very similar spatial pattern as in DJF (Figure 2e, f) and the annual bias (Figure 2b, c) in both GA7.1N and MERRA-2. The magnitude of the bias in MAM is approximately 1/2 of the magnitude in DJF due to lower solar insolation. We think one can therefore reasonably expect the nature of the underlying cloud bias to be similar in DJF and MAM (P13L19-22). Because the focus of the work is on improving the SW radiation bias, we think it is suitable to focus primarily on the months with the greatest bias (DJF) (P19L14-19).

**Since Ship-based measurements are also available in autumn and November-December, it would be very welcome to have also biases like the ones shown in Fig. 2 for the autumn and other summer months. Is the TOA SW bias spatial pattern (and the related comparisons between models) the same during these other months? I suggest Figure 3 to show only biases (the subplots m-p) for summer and autumn. The other maps (a-l) are difficult to read with the blue-shaded colourscale and I am not convinced they need to remain.**

We are have not analysed the November data (AA15 and HMNZS Wellington) to keep focus on the season with the greatest bias (DJF). We also wouldn't have data to cover a substantial part of the September-November (SON) season. The updated Figure 3 addresses the other points.

**Better discussing the cloud cover results in Summer/Fall (Figure 5) in relation to radiation biases in Summer/Fall would improve the overall discussion and conclusions of the paper. This would allow to make the most of the ship-based observations.**

This is now discussed in P19L14-16, P20L4-L20, and by adding a back-of-the-envelope calculation (Table 3; P23L9-14).

**Figure 5 is a great one and it would deserve better discussion in light of the motivations (i.e. the TOA SW biases over the SO and why theses biases are present).**

We have added more discussion in P15L26-P16L3 and P20L14-20. We think the relationship between the results shown in Figure 5 and the SW bias is relatively simple. Because Figure 5 shows the results based on the cloud mask, and the models underestimate the amount of cloud (as detected by the cloud mask), this leads to a proportional underestimate of outgoing SW radiation, unless the albedo of the cloud is overestimated (which is not analysed in our work). The SW radiation bias results in GA7.1N south of 60°S appear to be in line with this (Table 3), unlike MERRA-2 which is overestimating the outgoing radiation,

necessarily by overestimating the cloud albedo (P20L4-13).

We think it is useful to focus on the cloud amount and albedo, as it is important to fix the SW bias for the right physical reasons, i.e. the models simulate the correct amount and type of cloud at the correct altitude, and this is where our work makes a contribution. Other studies have already focused extensively on the analysis of TOA and surface radiation balance and the cloud albedo (through analysing the cloud phase). Our work is therefore complementary to these studies, and this is the advatage of ceilometers is over other instruments, but their ability to tell anything about the phase or the reflected SW radiation is limited.

We are currently working on a follow-up study which will focus more on comparing the individual cloud features observed by lidars and simulated by models. Our preliminary results suggest that GA7.1N is almost unable to simulate observed layers of stratocumulus cloud in the region and fog is either missing in the model or greatly underestimated. Therefore, the boundary layer, convection and large-cloud schemes will likely need to be fixed to be able to generate these types of cloud in the conditions typical in the SO.

**The authors note that GA7.1 reduces the SO SW radiation bias (e.g. in the abstract p1-L9). Figure 5 does not show any cloud from GA7.1 (only GA7.0). Why GA7.1 is performing better? Is this really because of better cloud representation (but we cannot see it from Figure 5)? And if not, what does it say about cloud being the main/only reason of SW radiation bias?**

We do not include GA7.0 in the manuscript any more (largely because the available run wasn't nudged, but also to limit the scope), but the updated Figure 5 shows that GA7.1 is much better than GA7.0 when it comes to simulation of cloud occurrence, and is now also better than MERRA-2. This should also explain much of the improvement in SW radiation in GA7.1 relative to GA7.0.

**Related to 1), what ERAI would give in Figure 3? More like the GCMs or like MERRA2? Perhaps ERAI brings this contrasting behavior that the authors highlight between GCMs and MERRA2, and this would allow to have both observations and reanalyses (only) used over the same period (2015-2018)**

While we agree that including ERA5 (ERAI does not contain the necessary fields) would be intereting, it would significantly expand the scope. The comparison with MERRA-2 already shows some interesting contrasting results, such as the different latitude of the split between the positive and negative SW radiation bias, the overestimation of cloud albedo in MERRA-2 (P20L4-13) and the cloud pase differrences (Figure 9). Perhaps a future study could focus on comparison of different reanalyses (ERA5, MERRA-2, JRA-55) with lidar observations in the region. Reanalyses are a suitable target for this kind of comparison due to having correspondence to the observed weather, while results from GCMs are often not available in a nudged mode, at least not via public repositories such as CMIP5, with the exception of the TAMIP project (GCM hindcasts). We therefore prefer to keep the scope limited to GA7.1 and MERRA-2.

**3)**
**The authors tend too often to rely on previous conclusions from previous papers (e.g. the Bodas-Salcedo et al. ones) to comment on what they find, rather than more thoroughly commenting/discussing their own novel results. The discussion part for instance gives much room to results of previous published study and/or to speculative comments about why GA7.1 is doing better than GA7.0 etc. and how MERRA2 is overcompensating for the low cloud-cover etc. Several sentences using "we cannot conclude. . .", "we cannot make the same conclusion. . . but it seems plausible. . ." "we cannot make substantial conclusions. . ." considerably weaken the discussion (section 4) from the beginning, and hence the paper, while it seems that all the ship-based measurements bring very valuable results (Figure 5, Figure 7, Figure 8) and interesting comparisons to MERRA2, and GCM runs (but cf. my point 1. on the time periods used).**

We have extended the Discussion with a more detailed discussion of the results (P19L30-P20L20, P20L30-P21L10, P22L34-P23L14). We explain in P20L4-13 why the statement about MERRA-2 overestimating the cloud albedo in not speculative but factual based on the results.

**4)**

**The discussion on the effect of sea ice is overlooked while it seems that some discussion could be made from Figure 8 (q,r and w,x). Also, while it seems that 8w is still showing some correlation, Figure 8x shows very different behavior and no attempt is made to comment on this. Given that a lot of the soundings you use (65) were made in 100% sea ice regions, and many of your CBH observations as well (I see the number of points present in your Figure 8x compared to the other similar subplots), I would expect to see more in-depth study of these observations, and this is really missing the present version of the paper in my opinion. For instance, the recent study by Jolly et al. (2018) that you cite showed the influence of different regimes on cloud cover: can the observation in Figure 8x be explain by particular synoptic-scale regimes or just by the sea ice being 100%? And why? Also, that other recent study by Listowski et al. (2018) that you also cite showed that not all low-level clouds anticorrelate with sea ice fraction but only the liquid-bearing ones. Can the behaviour you see in Figure 8x also be explained by clouds being of different sort/phase? The absence of correlation between CBH and min(SLL and LCL) may lead to think that you could be observing clouds advected from other places not related to local atmospheric conditions, that may be different in nature/phase from clouds over open water (you mention some hints towards the detection of supercooled liquid water with one of your instrument, can't you improve the Figure 8 by adding information on the phase, notably for Figure 8x?) In other words, can what you are observing from regions with 100% sea ice be explained by changing synoptic scale regimes, or cloud phase, or other things? Speaking of the sea ice regions, in Figure 7 the very low CBH are identified as being due to fog/very low clouds (p13-L20) and these are the points we also see in Figure 8x. Could this be blowing snow since we are in a 100% sea ice-covered region where snow can accumulate? In relation to cloud phase, in Figure 9 you compare LWP and IWP for GA and MERRA only for a specific year and month (jan. 2007). This does not seem satisfying to conclude for the longer time scales/other periods. Here again the use of different time-periods in the paper is not very welcome (see my point 1.). Do you really need Figure 9? If you really want to go into the cloud phase, using the lidar observations to assess the nature of cloud phase would be welcome. Or, as suggested in 1), perhaps only using MERRA2, ERAI (to contrast with MERRA2?) over 2015-2018 (only) would be a better option rather than using 2007, i.e almost a decade before the 2015-2018 ship-based observations. . .**

We do not discuss the effect of sea ice largely because cloud representation over sea ice makes little difference to the SW radiation bias (the surface is already highly reflective in SW). We consider the results interesting, but outside of the scope of the paper. But, this is potentially a topic for future effort that is being explored in our group.

The 65 radiosonde observations in Figure 8x were likely performed in a similar location in the high-latitude Ross Sea region (70–75°S). This region is likely affected by its proximity to land and not very representative of SO in general. These observations were only marginally related to our focus on SO.

We have updated Figure 9 to show a longer time period, which also better matches with the observations (DJF 2017/2018). We think this figure demonstrates nicely where some of the SW radiation difference between the models comes from. In our analysis we didn't use lidar observations which would be able to distinguish liquid and ice clouds. From one of the voyages we have data from a dual-polarisation lidar MiniMPL (but not from sea ice), which could allow us to perform such an analysis of cloud phase in the future, or provide the data to someone else upon request.

The radiosonde observations on TAN1802 were all performed in ice-free regions, and relatively far from any ice covered regions. Therefore, it is unlikely that the clouds could be advected from ice covered regions. None of the points in Figure 8x (70–75°S) appear in Figure 7 (60–70°S). This choice was made due to the likely effect of land/sea ice on observations performed between 70–75°S, while we intended Figure 7 to represent conditions in the open ocean.

**5)**

**Finally, the authors say that subgrid-scale processes should likely be responsible for the cloud mis-representation in models rather than the boundary layer thermodynamics but it is never said and**

**commented on what these subgrid-scale processes are. Do you mean the microphysics? Other processes? A discussion of what is used in the models regarding these processes would perhaps help to understand what should be improved in priority in the models and why the models are wrong. Using the contrasting behaviors of models to try to pin down the cause of cloud misrepresentation is an interesting method but the authors should provide with more clues in the discussion about what those subgrid-scale processes are and try to spot the main differences in the way the models implement these processes.**

We have added a paragraph in Discussion commenting on which subgrid-scale processes might be responsible (P22L34-P23L14). However, precise identification of the processes (we mean cloud parametrisation, boundary layer parametrisation and convection parametrisation in the Unified Model) is not something we can do without a more extensive analysis and especially running model experiments with modified parametrisation. Microphysics would likely have an effect on the cloud phase and thus SW reflectivity of the clouds, but we consider the cloud occurrence/cover a potentially larger problem than cloud albedo (see the added "back-of-the-envelope" calculation). We plan to focus on this problem in an upcoming paper.

————————————- **Line by line comments:** ————————————-
————————- **Abstract** ————————-
**P1-L9 By how much GA7.1 reduces the bias?**

We have removed GA7.0 from the analysis and only compare GA7.1N and MERRA-2.

**P1-L17 The analysis you mention is referring to your Figure 9 and the related comments. They only refer to the period January 2007. . . as mentioned in my major comment 4) this is not satisfying I think. When one reads the abstract it seems that you compare modelling and MERRA2 over the same period as the ship-based measurements, which is not the case. This is misleading.**

In the updated analysis we are comparing the same time period in the models and observations.

————————- **1.Introduction** ————————-
**P3 – L12 : "It was also more. . ." : what does "it" refer to exactly ?**

Replaced by "The clouds were also" (P3L18).

**P3 – L14 : "more likely to have intermediate cloud fraction" This is not clear. What is meant here by "intermediate cloud fraction"?**

"Intermediate cloud fraction" comes directly from Protat et al. (2017). We have clarified by adding "rather than very low or very high cloud fraction" (P3L18-20).

**P3 – L19-20 Please double-check and be more precise here (what "tuning" do you mean?). Kay et al. (2016) changed the threshold temperature below which detrained condensates are ice crystals and not liquid any more. They lower this threshold, allowing for more condensates to remain in the supercooled liquid phase when being detrained. The way the sentence is written suggests that ice crystals only are detrained.**

We have replaced "tuning" with "decreasing" (P3L25).

**P3 – L25 The reference to Jakob (2003) is a bit short or can be removed unless you specify what you mean by "cloud evaluation" regarding this specific study.**

We have removed the sentence.

**P3 – L27-35 Please make a new paragraph and give section numbers to help the reader.**

We have split the paragraph into two and added section numbers (P3L34-P4L8).

—————- 2. Method —————

**General comment: I would suggest a section 2. Datasets and 3. Method (lidar simulator). As it stands, it seems that this section combines too many different information about the data/methods used in the paper.**

We have split Methods into Datasets, Methods and Spatiotemporal subsets investigated (P4L9-P13L10).

**P4 – L2 As mentioned in my major comments, adding ERAI would be very interesting since this is a widely used reanalysis by the community, and would allow to contrast MERRA2 on same time-periods than ship-based obs.**

Please see the our comments above regarding this point.

**P4 – L9 I wonder whether a small appendix summarizing the main aspects of the lidar simulator would not be needed here, since the reference put is a paper in prep.**

We have added a reference to the website containing technical documentation of the new simulator (https://alcf-lidar.github.io) (P10L25).

**P4 – L16 What is the difference between GA7.0 and GA7.1? This would help understand and discuss the better performance of the latter in terms of SW bias (as stated in the abstract).**

We have removed GA7.0 from the analysis and focus solely on GA7.1N and MERRA-2.

**P4 – L 18 As said in the major comment it should be explained why these runs are used. 1980-1989, and then 2007. Why not having runs over more recent periods (as the ship-based measurements).**

We have removed this section. Only one GA model run and one time period are investigated in the up-dated analysis.

**P4 24 – "Can only be compared statistically" What do you mean? Please clarify.**

We have removed this sentence (GA7.0U is no longer used). The nudged run GA7.1N is now compared 1:1 to observations.

**P4 – L26 "Limited data availability. . ." What do you mean? See my major comment 1) It does not seem that you are saying over which period you analyse MERRA2. This should appear in this section.**

We have removed the sentence on "Limited data availability" (addressed by using a nudged GA7.1). We have added a sentence to the MERRA-2 section regarding the analysed time period (P8L16), which is the same as the period of observations.

**P5 – L15 "downsampled" from what initial resolution?**

In the updated analysis we used the original MERRA-2 resolution without downsampling. We have re-moved the sentence.

**P6 – L5 "appears largely zonally symmetric" I don't think we can say the pattern of the bias is symmetric, even zonally, but rather that the bias is present across all longitudes, but its magnitude does change zonally.**

We have replaced the statement with "the SO SW radiation bias is present at all longitudes in the SO"

(P6L14-17).

**P6 – L6 "with a notable exception. . ." Precisions not needed in the section presenting the ship measurements. . .**

We have replaced this part by "SO SW radiation bias is present at all longitudes in the SO (...), affected by atmospheric circulation in the SO (...)." (P6L14-17).

**P6 – L8 "Figure 1. . ." This Figure is already mentioned before P5 – L21.**

We have removed the sentence.

**P6 – L20 I am not sure what is meant by "directly reveal the cloud liquid. . .". The strength of using a simulator is to compare the observables and not to rely on all the hypotheses used by inversion routines to retrieve IWC and LWC from lidar observations.**

We have replaced the sentence by: "Due to signal attenuation and noise ceilometers cannot measure clouds obscured by a lower cloud, and therefore cannot be used for 1:1 comparison with model clouds without using a lidar simulator, which accounts for this effect (Chepfer et al., 2008)." (P6L29-P7L3).

**P7 L24 – Please clarify the title e.g. "Geographical areas/domains investigated" or "Domains used for the analysis" Also, having 2.1 as "Datasets" then 2.2 as "Domains" then 2.3 "COSP simulator" is not ideal I think, and I would first present all datasets and tools, and then the domains.**

We have changed the title to "Spatiotemporal subsets investigated" (P12L19). We have also restructured the text as suggested. Subsections of Methods are now split into Datasets, Methods and Spatiotemporal subsets investigated (P4L9-P13L10).

**P8 – L12 The title of this subsection is misleading since you are not using COSP simulator in the end, but your own simulator. Please change the title accordingly.**

We have renamed the section to "Lidar simulator" (P10L1).

**It seems to me you don't need a section 2.3 and you could have everything put in current 2.4.3 where you could at once explain the modeling of the lidar signal along with its processing.**

Some of the same lidar processing steps (such as cloud detection) are applied on the simulated lidar as on the observations. Therefore, the current arrangement make sense from this perspective.

**P9 L15 What is this known value of LR? Where does it come from?**

The value is 18.8 ± 0.8 sr as stated in the paper referenced in the paragraph (O'Connor et al., 2004). We have added this number in parentheses (P11L14).

**P9 L21-22 Citing Kotthaus et al. at the end of the paragraph falls a bit short and I am wondering if it should not appear earlier in the paragraph with some more explanation about why you refer to this study. Are you using their method? Then please say it.**

We have removed the sentence.

**P10 L7-9 Why do you need to do this? How are these random samples used then?**

Added sentences "The lidar simulator processes each sample individually. The resulting cloud occurrence is calculated as the average of the 10 samples." (P12L7-9).

**To shorten this section 2. I would not define SLL here, rather when it is used for the first time. Also SSL is neither a dataset, nor a tool, rather a variable defined to help with the analysis. Also, is there any past reference using this definition? If yes, please cite relevant paper.**

We have relocated the paragraph introducing SLL to the Results section (Section 5.3) (P16L31-P17L7). The authors are not aware of any references of previous use of SLL or an equivalent metric. We noted the relationship between SLL and CBH as theoretically plausible and later confirmed by joint radiosonde and ceilometer observations (Figure 7).

—————————- 3. Results —————————-

**P10-L26 to P11 L-16 There are too many statements dealing with observations made on Figure 3 that are actually difficult to see, whereas Figure 4, introduced after, is more helpful to confirm statements made by the authors. Also, as suggested in major comments, I would tend to simplify Figure 3 by showing only the biases and remove all the blue-shaded figures where the biases are difficult to read, especially regarding the statements made by the authors in the main text.**

We have updated Figure 3 to show biases, which should now be more clear.

**P10 – L28 "Lower" than what? And what biases? Please clarify.**

The biases are discussed in Loeb et al., 2018 referenced in the sentence. We have clarified in parentheses (P13L16).

**P10 – L29-30 I would remove the sentence about the "predominantly zonally symmetric pattern" and the "more variable patterns in the tropics", which is not very clear to me.**

We have replaced the part with "relatively zonally symmetric pattern of negative and positive bias" (P13L17-19), as the updated figure now shows the bias more clearly, and removed the part about the tropics, which are not covered in the updated figure.

**P11 – L1 "upwelling and downwelling" what?**

We have removed the sentence. Figure 3 is now zoomed on the SO rather than covering the tropics to better highlight the features.

**P11 – L3 "large differences" between what? I would drop the mentions to the Peninsula and what is happening to the east of it as it is not clear why one would give so much importance to this since the ships did not get there anyway.**

"Large differences" between the models and CERES. We have removed this part of the sentence (P13L23-25).

**P11 – L1 I don't understand the footnote. Also, I am not convinced there is a need to highlight a particular day in the present paper.**

We have removed the footnote. The day picked in Figure 3 and Figure 9 is now 1 January 2018. We think that it makes sense to keep the daily plots due to the stark difference in TOA outgoing SW radiation between CERES and the models visible on the daily means, consistent with the statistical results. We have changed the scale and colormap of the plots to better highlight the difference.

**P11 – L4 One cannot really see this "greater reflectivity".**

This should be visible in the updated Figure 3. We have replaced the sentence with "The region on the eastern side of the Antarctic Peninsula shows the greatest negative bias in the models (Figure 3b, c, e, f)". (P13L25-27).

**P11 – L13 "With some individual cloud systems being too bright". I am not sure this should remain in the text. Again, I think all the consideration about the blue-shaded maps in Figure 3 (but biases maps should be kept) should be removed and Figure 4 should be used instead.**

We have replaced the blue-white colormap with a grayscale colormap on a smaller scale of values and smaller span of latitudes. The differences between the models and CERES in the updated Figure 3 should be more obvious now. We have also removed the part of the sentence "with some individual cloud systems being too bright" (P14L2-6).

**P11 – L21 "cyclical". Rather "seasonal"?**

Replaced with "seasonal" (P14L13, P14L16).

**P11 – L20 What is meant by "likely a secondary modulating factor". Please be more explicit. A modulating factor for what?**

Modulating factor for SW radiation. We have clarified: "modulating factor of the TOA outgoing SW radiation" (P14L14-16).

**P11- L26 "These panels also justify why. . ." Not needed.**

We have removed the sentence.

**P12 L4-6: The two sentences fall a bit short. Also, they would be in better place in the discussion part, with more explanations. ". . ..in the GA7.1 model": so what?**

We have removed this paragraph (we no longer compare GA7.0 and GA7.1).

**P12 L2 Figure 5 is very interesting and rich, and more analysis should be provided also regarding similarities or differences between summer results and autumns results. (Please consider adding letter to designate specific subplots of Figure 5). Also it seems that obs and model agree more where the statistics is larger (more days), can't you say something about that? Isn't it possible that at other time/places the larger disagreement between model/obs is partly due to smaller statistics of observations? This relates to my major comments that more analysis and discussion are really needed on this plot.**

The results from the updated Figure 3 suggest that there is little difference in the TOA outgoing SW radiation bias between the austral summer (DJF) and autumn (MAM) (the geographical pattern is very similar, except for the magnitude modulated by the incoming solar radiation). We therefore expect the bias has the same underlying cause in both DJF and MAM. MAM is also much less important in terms of fixing the SW bias in models due to much lower solar insolation in the season. Also Figure 5 does not indicate that there is a significant difference in cloud occurrence between DJF and MAM.

We think that greater number of days in Figure 5 does not necessary imply much greater weight of the result due to time correlation of weather patterns and correlation with sea ice concentration around Antarctica. Therefore, we partially consider the subplots of Figure 5 as independent "snapshots" each with the same weight (solid lines in Figure 6), in addition to calculating the weighted averages (dashed lines in Figure 6).

We have added labels to Figure 5.

**P12-L10 What period is used for MERRA2 here?**

This has been addressed by removing the statistical comparison with GA7.0U and stating the time period in Methods (P8L16).

**P12 L12. As mentioned in the major comment. Why can the authors trust comparisons between simulation of the 1980s period and the 2015-2018. This should be much better introduced/justified.**

We have removed the sentence (addressed by using a nudged model).

**P12 L19-20 how much higher?**

We have removed this sentence. Schuddeboom et al. (2018) evaluated GA7.0 which had a much greater bias than now evaluated GA7.1.

**P12 L20-21 "Due to the zonal. . .of the whole SO" Could suit the discussion part. Not needed here.**

We have removed the sentence.

**P12 L27-34 I would drop Figure 6 and give only numbers. It saves a Figure.**

We would like to keep Figure 6 as it gives a good visual summary of Figure 5. We are also using the numbers derived from this Figure in the abstract, and in the "back-of-the-envelope" calculation added in the revised manuscript.

**Also what bothers me is that GA7.1 is said to be better from nudged simulations but, in the end, only GA7.0 is presented here, because of the decadal run being only available with GA7.0. This is again a shortcoming of accepting to work with so many different time periods for different simulations.**

This has been addressed by using GA7.1N/2015–2018 instead of GA7.0U/1980-1990.

**P13 – L1-5 Have the authors consider to use satellite data, or to rely on previous publications to try to assess how the comparison to models is biased by extinction of the ceilometer signal into the lowest thick clouds? At least this should be discussed in the discussion part. This is not the case now.**

The lidar simulator accounts for signal attenuation. Therefore, the comparison with the models is not biased by extinction.

**P13 L10-11 Is the extraction made above the lat/lon of the balloon launch or does it follow the radiosonde trajectory? I guess it is the latter but you may want to clarify this in the text.**

It is the former. The resolution of the models is generally not high enough to make a difference. The balloon trajectory length was on average 58 km on the TAN1802 voyage, and the higher altitudes when the balloon was further away from the ship would likely not affect the analysis, which mostly found differences in clouds in the lowest part of the troposphere. We have clarified the text (P16L23-30).

**P13L11 Can you make a subplot for each of the dataset? It is difficult like this to spot differing behaviours between coloured markers.**

We have increased the size of the markers to make them easier to distinguish.

**P13 – L14 What relationship?**

Added "observed and modelled relationship" (P17L11-12). CBH and min{SLL,LCL} and the axes of the scatter plot. The relationship is between these two coordinates of the points.

**P13 – L19 How large?**

We have added quantification in parentheses (P17L12-14).

**P13- L23 how weaker?**

This is now quantified by adding two subplots in Figure 7 for SLL (Figure 7c) and LCL (Figure 7d). We have added text "weaker relationship than min{SLL,LCL}: 26% and 31% of observed profiles have CBH within 100 m of SLL and LCL, respectively (Figure 7c, d)." (P17L19-20).

**P13 – L25-27 The fact that LTS is not a good indicator should be discussed in the discussion part and I don't think it is the case for now. This relates to my major comment 3) where I suggest that more emphasis should be given in the discussion to all results obtained from these novel ship-based measurements.**

We think Figure 7 demonstrates relatively well why min{SLL,LCL} is a better predictor for CBH than LTS. The correlation coefficient in OBS for min{SLL,LCL} (Figure 7a) is 0.4, and for LTS (Figure 7b) -0.2. The main difference, however, is that min{SLL,LCL} is very close to CBH (within 100 m) in 40% of cases. LTS cannot be used as a 1:1 predictor for CBH due to having different units (K vs. m), and the correlation is not strong enough for a linear model. The updated Figure 7 now also shows the graphs based SLL and LCL as predictors, both of which show an inferior relationship with CBH.

**P13 – L28-34. Figure 8 is introduced, but then some general statement are made about synoptic scale forcing. It would be much better, for the reader, to stick to the Figure.**

The general statement explains why we are showing Figure 8. Therefore, we would like to keep it.

**P14 – L1 "As can be seen. . .where there is no sea ice". What can be said about Figure8a and b where there is no cloud but at the same time GA7.0 is not in agreement at all with observations? Also what is the unit in the x-axis of subplots Figure 8a-f and Figure 8m-r? In Figure 8g-l and s-x, you are not showing the modelled dots, only observations. I would have expected to see the model outputs as well. Or is it not useful here?**

Figure 8a, b now show much better agreement with GA7.1N (as opposed to GA7.0U), most likely due to the nudging. We have added units to the axes.

We have removed the scatter plots in Figure 8 (Figure 9 shows similar information more clearly).

**P14L3-4 "There is no substantial difference between. . ." This is not true for Figure 8a and b. . . which present non-sea ice cases. This should be discussed. "Plausible effect"? What do you mean?**

GA7.1N now shows a much better agreement in Figure 8.

By "plausible effect" we mean the theoretical expectation of how min{SLL,LCL} relates to convection as explained in the introduction of the quantity. We have removed this part of the sentence (P18L3-6).

**P14 L8 – What is meant by subgrid-scale processes? Please be more specific.**

We have commented on the subgrid-scale processes by adding a paragraph in Discussion (P23L4-14).

**P14 - L2 I am not sure about this subsection. I struggle with having it only focusing on January 2007. Since the novelty of the paper is the ship-based measurements I am not sure having this part here is relevant, especially that it is only about comparing Jan 2007 for two models. Plus, the GA7.0 one is not the one used in Figure 5, but the nudged one, and it is not clear what period is used for MERRA2. Why not showing also GA7.1 since it is spotted as reducing the SW biases (because of the modelling of larger supercooled LWP?)**

This has been partially remedied by analysing all models for the same time period as the observations.

In the updated analysis GA7.1 shows much better match with observations in terms of cloud occurrence (Figure 5). Therefore, it is expected that the improvement of TOA outgoing SW radiation bias over GA7.0 can be largely attributed to the improvement of cloud cover representation rather than improved super-cooled LWP.

**P14L26-30 "We should note. . ." These are comments for the discussion part, but even so, these considerations are also and already mentioned in other places of the paper and remain very general and a bit speculative. I am not sure these zonal plots deserve a separate section, also because of these time period issues mentioned above.**

We have removed the statement.

The zonal plots now show the same time period as the rest of the analysis.

**———————— 4. Discussion ————————**
**In general the discussion should be more focused on your results at least in the beginning and spend less time on explaining previous works. Figure 10 (which is interesting indeed) should come earlier in the discussion. Also, you don't seem to do discuss Figure 10b, but only Figure 10a. Sentences like (P15-L18-21) "Combined. . ." are a bit speculative and more room should be rather given to discussing the results obtained from ship-based measurements, ie. Figure 5, Figure 7 and Figure 8, and 10. And then make the link to the TOA SW bias issue and relate it, possibly, to the LWP as modelled (cf. Figure 9 – if still considered relevant in a revised version).**

We have relocated introduction of Figure 10 to Results (P15L1-7) and replaced Figure 10b with equivalent plot based on MERRA-2.

The statement P15-L18-21 (in the original manuscript) was not speculative in the context of the results – we have shown that the cloud cover is underestimated in MERRA-2 (Figure 5), and the only way the model can overestimate the total (all-sky) TOA outgoing SW radiation (Figure 3, 4) at the affected latitudes and time of year is by overestimating cloud albedo. We have clarified this point in the Discussion (P20L4-13).

**P15-L34 to P16 L4. This is too much about other study, not enough discussing your results. Figure 10 comes after that and this is not appropriate. Also – as an example of additional discussion element – are the ship-based observations, which show larger discrepancies from MERRA2, in places where the near-surface temperature is the coldest? In other words, can you relate Figure 10 with your cloud results, instead of only speaking of the SW bias?Also, why is Figure 10 only showing the year 2007? Why not showing the decadal simulation, and the MERRA2 outputs as well (during the ship-based measurements)? What do they say? How is it consistent or not with the cloud simulations in these models? These sorts of analysis/discussions are really missing in the paper, in my opinion.**

We have added a subplot showing MERRA-2 and extended the time period to January 2018. We have relocated introduction of Figure 10 to Results (P15L1-7).

**P17 L9 "Because sea ice is an important factor. . .": What is meant by "secondary effect on cloud cover"? It seems to me you have the opportunity to say something about the effect of sea ice on very low clouds (and specifically the ones missed by satellites) – e.g. your Figure 8x – but you are not exploring this in the paper. This goes along with my major comments that not enough efforts are made to discuss the very interesting observations you have from ship over three years and in sea-ice free/covered regions.**

The focus of the paper is on improving SW radiation bias in GCMs. Even though the difference between ice-free and sea ice cases is interesting, clouds over sea ice covered regions have relatively small impact

on the SW radiation (the ice covered surface is already highly reflective, and presence or absence of clouds makes little difference). Therefore we prefer to limit the scope of the paper mostly on the ice-free regions. This is an area that might be completed in future studies within the group.

**————————— 5. Conclusion —————————**

**In the conclusion only you speak again about the subgrid-scale processes without specifying them. This should be a paragraph on its own in the discussion part, trying at least to understand how the various models are doing different in parameterising these processes. This would give more perspective to the present work I think.**

We have added a paragraph in Discussion commenting on this problem (P22L34-P23L8).

**—————————- Figures. —————————-**

**Figure 2 If you still want to keep all the model results (provided you better justify your method – see my major comments) then you should add the time-periods for the simulations you use, and for the observations, so that one immediately knows you are using different times for comparisons (and that this is then discussed in the text).**

This has been addressed by leaving out GA7.0U/1980-1990 and instead comparing GA7.1N, MERRA-2 and observations over the same time period.

**Figure 3 As I said before, one struggles to see features with a single colour-shaded scale. As suggested I would keep only the plots showing the biases, and for summer and autumn (as these are seasons investigated with ship measurements).**

We now show bias in DJF and MAM (2015–2018) with a latitude range of 45–75°S. We have changed the colormap, which should better highlight the differences.

**Figure 4 The horizontal line indicates the "0" value for the bias (red curve). Please make it red (and thicker, or dashed).**

We have made the line dashed, thicker and red.

**Figure 5 What period is used for MERRA2? Why not also showing the nudged runs with the better (according to what you say) version GA7.1U.**

This has been addressed by using GA7.1 nudged for 2015–2018. GA7.1U was not available in our original analysis.

**Figure 6 Not sure this figure is needed. See my comment in the relevant section.**

We use results from this figure in the abstract to quantify the cloud cover bias, and to perform a "back-of-the-envelope" calculation added to the revised manuscript. Therefore, we think the figure adds valuable summary information.

**Figure 7 This would be better to separate the dataset in different subplots to see the different behaviours.**

We have increased the size of the markers to make them easier to distinguish.

**Figure 8 What are the x-axis units in the subplots a-f and m-r? The markers in the g-l and s-x subplots are quite small. Can you either make them larger or increase the size of the subplots.**

We have added x-axis units and increased scatter plot markers.

**Figure9 What are the contour values?**

We have added contour level values.

**Anonymous Referee #2**

**Review Kuma et al: 'Evaluation of Southern Ocean cloud in the HadGEM3 general circulation model and MERRA-2 reanalysis using ship-based observations' ( MS No.: acp-2019-201) The authors conducted analysis of three model datasets by focusing on the Southern Ocean to understand errors in models in the shortwave (SW) radiative flux at the top-of-the-atmosphere, using ship observational dataset as well as satellite observations to understand the errors. They found that GA7 runs and MERRA-2 runs have the opposite bias in the outgoing SW flux (underestimate in GA7, overestimate in MERRA-2) over the southward latitude of 55S. They compared their cloud amounts with the ship observations and showed that both models underestimate their cloud amounts. They also conducted nudged-runs and showed that there is a big difference in cloud liquid water amount in these models, concluded that the main source of the difference in their SW bias is from the difference in their cloud properties, which are determined by the sub-grid cloud parameterizations. The shortwave bias over the Southern Ocean tends to be a common problem in climate models. This is a nice piece of work which contributes to improve our understanding of the representations of clouds over the region. However, current manuscript misses some information for their logic to convince readers, hence the key message remains unclear. I suggest this paper to be published after a minor revision.**

**Main comments: Although GA7 runs and MERRA-2 runs have the opposite bias in the outgoing SW flux over the southward latitude of 55S, both HadGEM3 GA7 and MERRA 2 underestimate cloud amount. In Discussion section, the authors mentioned that models may fail to represent fog or low cloud which are generated by convection which are induced from subzero airmass from polar regions over warm water. What our community is keen to know is whether we can improve the representations of such clouds in GCM or we should seriously start thinking of using cloud resolving model or GCM. Whether/how much the underestimate of the cloud amount improves in their nudged runs will provide a clue for it. The authors should add a figure which shows cloud amounts in free run and nudged runs.**

To address the major comments of Referee 1, we have replaced the free running model with a nudged model (GA7.1N), and Figure 5 now compares the nudged model, MERRA-2 and observations. Based on the updated results, we think it is possible to fix the parametrisation schemes to generate more low cloud (of the right type) and fog in the conditions typical in the region. GA7.1N is underestimating the cloud cover by about 4–9%, but this is still enough to cause a relatively large bias in SW radiation (Table 3).

We are currently performing a more detailed analysis of the lidar backscatter vs. a nudged model with the aim of getting the model to simulate the missing types of cloud, especially layers of stratocumulus and fog on certain days. This should be the subject of a paper in preparation. We think it is possible to fix the parametrisation schemes in GCMs, which have likely been tuned for other regions globally and neglected the SO due to the lack of ground-based observations (satellite observations do not identify these types of cloud correctly if obscured by a higher-level cloud).

**The authors showed that main difference in SW radiative flux bias over the Southern Ocean between HadGEM3 GA7 runs and MERRA 2 runs is cloud water amount. This shows a big impact of subgrid cloud parameterizations on radiation. Please check subgrid cloud parameterizations in GA7 and MERRA2 then discuss which parameterization could potentially cause the difference in radiative flux. Since the authors showed the opposing sign of the SW CRE south and north of 55S in GA7.1, it would be useful to apply the same analysis (comparison to the ship observations, analysis of the nudged runs) to the region of the north of 55S, confirm whether the smaller error is because**

**of the (less worse) representations of the cloud amount over the region.**

We have added a paragraph in Discussion which comments on the possible subgrid-scale parametrisation schemes in GA7.1N responsible for the bias (P23L4-8). A concrete identification of the problem will likely require experimenting with the parametrisation schemes to achieve a better match with the observed cloud occurrence profiles. The observed SW radiation bias is likely a combined effect of underestimation of cloud cover and overestimation of cloud albedo, resulting in the latitudinal gradient of bias, which is positive north of about 55°S (65°S) in GA7.1N (MERRA-2) and negative south of this latitude.

We already compare a large number of geographical subsets (5x6) and therefore by adding more we would make the plots even more complicated. The latitude of 55–60°S already covers a region of the positive bias in GA7.1N in the Ross Sea sector (Figure 3e). Figure 5a1, a2, a3 compare this region between the observations and GA7.1N and show that GA7.1N and observations had cloud cover of almost 100%, but with differences in fog/low cloud simulation (the model didn't simulate the correct altitude of cloud). Because the error in cloud cover in this region was so small, this points to cloud albedo overestimation as the reason for the positive SW radiation bias, even though Figure 5a1, a2, a3 are over a relatively small number of days (4.4).

**Minor comments: Discussion: the beginning (L1-10) was difficult to read, because the authors mention the opposing sign of the SW CRE south and north of 55S in GA7.1, but then solely talk about the results over the south of 55S.**

We have changed this part of Discussion also to address multiple comments of Referee 1 (P19L5-).

**Figure 6: Clarify what is the weight for the weighted average.**

We have clarified that the weight is the number of days the ship spent in the spatiotemporal subset.

**Figure 8: add grid values to the Frequency axis**

We have added units to the x-axis in Figure 8.

**P11-l1: 'upwelling and downwelling' Where are regions of upwelling and downwelling radiative flux? If the authors are talking about large scale circulation, these should be 'ascent and descent'.**

We have removed the part of the sentence (Figure 3 is now focused on the SO only) (P13L17-19).

**P11-l4: I cannot see the results described about models. And the contrast between western and eastern sides of the Antarctic Peninsula contradicts to the following description 'The zonal symmetry. . ..'**

We have updated Figure 3 to show biases and increased the contrast and scale of the plots. We have replaced "zonally symmetric" with "relatively zonally symmetric" (P13L17).

**P11-l14: Figure 3p?**
**P11-l32: 'consistently positive': negative in Sep-Dec in 60S-70S**
**P11-l33: 'also lower than GA7.0 and GA7.1': not necessarily in GA7.1**

We have updated this part to account for the updated Figure 3.

**P13-l14: Did you define SLL and LCL? (Super liquid level and lifting condensation level?) How did you define SLL?**

We use the traditional definition of the lifting condensation level. We now define the "SST lifting level" (SLL) in the Results section before the first use of the quantity (P16L31-P17L7).

**P13-l22: Give a speculation why min(SLL, LCL) is better correlated with CBH than SLL/LCL individually.**

We have added a section in Discussion which details why we think the relationship of CBH with min{SLL, LCL} is better than SLL/LCL (P20L30-P21L3).

**P14-l5: Provide a figure or reference about SLL in GA7.0 is higher than observed.**

In the updated Figure 8 we plot distribution of min{SLL,LCL} instead of SLL (to be more consistent with the rest of the analysis). The updated description of the figure comments on the observed and modelled distribution (P17L26-P18L10). GA7.1N represents the observed distribution relatively well.

**P14-l16: Fig 9. It is not clear why the authors create these plots over two different backgrounds.**

We prefer to show both fields due to their effect on cloud (potential temperature through convection and relative humidity through condensation).

**P14-l18: Fig 9. Not clear. Different colors should be used for different levels to show this.**

We have lowered the value of the lowest contour to 12 $\text{gm}^{-3}$, which means some cloud ice contours are now visible on the MERRA-2 plots.

**P14-l29: cloud cover a reduce ..': typo?**

We have removed this sentence due to the overall change of a dataset.

**Fig 5: Why did the authors exclude 50S-55S for the plots?**

We did not include 50–55°S in order to keep the paper relatively focused. The radiosonde observations on TAN1802 and NBP1704 voyages were only available south of 60°S, which limited some of the plots.

We agree that 50–55°S might be an interesting addition in Figure 5. We can extend this figure before a final revision of the manuscript if the referees think it useful in the updated analysis.

**Fig 8: The authors did not analyze model results in other latitudes where clouds shows the opposite bias (in 50S-55S).**

We could not provide plots in Figure 8 north of 60°S due to radiosonde observations only available south of this latitude.

**P15-l10-11: I cannot follow the logic here.**

We have added a paragraph in Discussion explaining this point (P20L4-13). The logic is that the effect of clouds on reflected SW radiation is the product of cloud cover (the cloudy fraction of the sky) and cloud albedo (reflectivity of the cloud). We have shown that cloud cover is underestimated in the models, while at the same time MERRA-2 overestimates the reflected SW radiation. Therefore, if the first factor of the product (cloud cover) is underestimated, the second factor (cloud albedo) must be overestimated to get overestimated reflected SW radiation.

**P16 l23: Is it possible to add the definitions of supercooled liquid in GA7.0 and MERRA-2?**

We consider any cloud liquid at air temperature below zero supercooled. We have clarified this at the first mention of the term in the Introduction (P3L8). We have also added a statement in Figure 9 caption clarifying that all cloud liquid in the plot is supercooled.

**P17 l11: Is this a result from the nudged run or from other studies?**

We have removed the sentence. In the updated analysis we are comparing with the nudged run which uses sea ice concentration prescribed from satellite observations.

[revised manuscript text omitted]

---

## Author Response (AR3)

**Response to referees on "Evaluation of Southern Ocean cloud in the HadGEM3 general circulation model and MERRA-2 reanalysis using ship-based observations" by Peter Kuma et al.**

Peter Kuma, Adrian J. McDonald, Olaf Morgenstern, Simon P. Alexander, John J. Cassano, Sally Garrett, Jamie Halla, Sean Hartery, Mike J. Harvey, Simon Parsons, Graeme Plank, Vidya Varma, Jonny Williams

April 12, 2020

We would like thank the referees for reviewing the revised manuscript. Please see our response to the minor comments below.

The page and line numbers refer to the new latexdiff document.

**Comments by Anonymous Referee #1**

**P3-L14: You should give a reference for AMPS: Powers, J. G., Manning, K. W., Bromwich, D. H., Cassano, J.J., and Cayette, A. M.: A decade of Antarctic science sup-port through AMPS, B. Am. Meteorol. Soc., 93, 1699–1712,https://doi.org/10.1175/BAMS-D-11-00186.1, 2012.**

We have added the suggested reference (P3-L6).

**P3-L22: the most recent study investigating SLW in the SO is the one by Listowski et al. 2019 I think, and it should appear here after Jolly et al. 2018: Listowski, C., Delanoë, J., Kirchgaessner, A., Lachlan-Cope, T., and King, J.: Antarctic clouds, supercooled liquid water and mixed phase, investigated with DARDAR: geographical and seasonal variations, Atmos. Chem. Phys., 19, 6771– 6808, https://doi.org/10.5194/acp-19-6771-2019, 2019.**

We have added the suggested reference (P3-L14).

**P6-L13: AA15 is not labelled in Figure 1. I guess it is AA V1-V3.**

We have changed the label in the figure to AA15 (P27).

**P6-L14-15: Please double-check the sentence: "...is present at all longitudes in the SO (section 5.1), affected by atmospheric circulation in the SO (...). ("in the SO" repeated twice)**

We have reforumulated the sentence (P4-L20).

**P7-L9: Define here GNSS (this is only done at L18 for now).**

Fixed (P5-L12, P5-L21).

**P15-L27: peaking below 500m (not "km")**

Fixed (P11-L29).

**P18-L2: there is no label b4, do you mean c4?**

We have fixed the figure labels ("c" -> "b") (P35).

**P18-L2-L3: This is not so clear or please explicit why you can say this. As it can possibly be said for (a1) and (a2), this is not so sure about the other sea-ice free cases.Please explain (by quantifying biases?) how you see in Figure 8 that MERRA-2 is worse than GA7.1N.**

We have retained only the part "is less likely to peak near the ground" and added reference to the figure subsets which show this deficiency (P13-L9).

**P18-L34: I am not sure to agree ("is majority liquid"). Looking at Figure 9i: IWP>LWP! P19-L3: is almost entirely liquid. Not ice (looking at Figure 9j)**

Fixed (P13-L25,26).

**P19-L9: in both models**

Fixed (P13-L31).

**P19-L14: similar**

Fixed (P14-L3).

**P20-L14: "Remarkably, the observed and simulated cloud occurrence profiles do not appear to be significantly different between the DJF and MAM seasons or different latitude bands between 55 and 70 °S (Figure 5)": I am not sure to agree with the authors, unless you specify what you mean by "significantly". Looking at all the profiles in Figure 5 there are clear differences (multiple layers, single low layer, etc.). So, I am not sure about the rest of the paragraph either ("This is in contrats..."). Please clarify and improve this part.**

We have reforumulated the text to be more explicit (P14-L25–33).

**P21-L6-7: Following my comment on P18-L2-L3 I am not sure to agree with this or this is not convincingly demonstrated.**

We have changed this to "underestimates cases when min{SLL,LCL} was near the surface" (P15-L23,24).

**P21-L29: Are you referring to cold air outbreaks here? They tend to form cloud streets, and one expects models to struggle with these low clouds. Any reference here to back up your comment?**

Yes, we are refering to cold-air outbrakes. We have added "cold-air outbreaks" in parentheses and added a reference to Bodas-Salcedo et al., 2012 who discuss model biases in this situation.

**P21-L35: Rewrite here the references used in the intro, after "...in summer months"**

Done (P16-L10).

**P22-L31: Please give at least one reference here.**

Unfortunately we couldn't find a good reference. We think this point is demonstrated by the fact that the nudged run of GA7.1 and the reanalysis take sea ice from observations. Therefore, it is unlikely that the radiation biases are due to misrepresentation of sea ice in the models. We have reformulated the sentence (P17-L1–3).

**P23-L15-30: Since you define a new quantity in your study (SLL), this would be interesting to recall it here, and explain the benefit of using it.**

We have added a paragraph in the conclusions discussing this new quantity (P18-L7–12).

**Figure 9: what are the contour labels? On subplots c, d, g, and h, one cannot see at all the different red contours. Please improve the figures, and indicate contour values (in the caption or in the plot).**

We have added units to the legend and increased the intervals between the contour levels. The contour values are indicated on the countour lines (P36).

**Comments by Patrick Chuang**

**I also acted as the second referee for the revised manuscript. I have only one comment which is that Fig. 10 appears out of order (the first reference is between the first references to Fig. 4 and 5 from what I can tell). Is that intentional? If not, please fix. Thanks.**

We kept in original order of figures in the second revision in order to make the discussion easier. I have fixed the numbering in the current revision.

[revised manuscript text omitted]